Macromammalian faunas, biochronology and palaeoecology of the early Pleistocene Main Quarry hominin-bearing deposits of the Drimolen Palaeocave System, South Africa

Adams Justin W. 1 justin.adams@monash.edu
Rovinsky Douglass S. 1
Herries Andy I.R. 2 3
Menter Colin G. 3
1 Department of Anatomy and Developmental Biology, Monash University , Melbourne, Victoria , Australia
2 The Australian Archaeomagnetism Laboratory, Department of Archaeology and History, La Trobe University , Bundoora, Victoria , Australia
3 Centre for Anthropological Research, University of Johannesburg , Johannesburg, Gauteng , South Africa
Jungers William
Electronic publication date: 2016 Apr 18
Publication date: 2016
Volume: 4
Electronic Location ID: e1941
Received 2016 Mar 6; Accepted 2016 Mar 25
Copyright: ©2016 Adams et al.
Copyright year: 2016
Copyright holder: Adams et al.
License: This is an open access article distributed under the terms of the Creative Commons Attribution License, which permits unrestricted use, distribution, reproduction and adaptation in any medium and for any purpose provided that it is properly attributed. For attribution, the original author(s), title, publication source (PeerJ) and either DOI or URL of the article must be cited.
License URL: https://creativecommons.org/licenses/by/4.0/

Keywords: Chasmaporthetes, Paranthropus, Megantereon, Homo, Dinofelis, Lycyaenops

Funding: National Research Foundation, South Africa (African Origins Platform Grant) Centre for Anthropological Research of the University of Johannesburg Australian Research Council Future Fellowship FT120100399 Monash University Funding for excavation and curation of the Drimolen Main Quarry deposits was provided by grants to CGM by the National Research Foundation, South Africa (African Origins Platform Grant) and by the Centre for Anthropological Research of the University of Johannesburg. Additional funding for excavation was provided by an Australian Research Council Future Fellowship (FT120100399) to AIRH. Funding for the faunal analysis presented here was provided by JWA through internal research development funds from Monash University. The funders had no role in study design, data collection and analysis, decision to publish, or preparation of the manuscript.

==============================
The Drimolen Palaeocave System Main Quarry deposits (DMQ) are some of the most prolific hominin and primate-bearing deposits in the Fossil Hominids of South Africa UNESCO World Heritage Site. Discovered in the 1990s, excavations into the DMQ have yielded a demographically diverse sample of Paranthropus robustus (including DNH 7, the most complete cranium of the species recovered to date), early Homo, Papio hamadryas robinsoni and Cercopithecoides williamsi. Alongside the hominin and primate sample is a diverse macromammalian assemblage, but prior publications have only provided a provisional species list and an analysis of the carnivores recovered prior to 2008. Here we present the first description and analysis of the non-primate macromammalian faunas from the DMQ, including all 826 taxonomically identifiable specimens catalogued from over two decades of excavation. We also provide a biochronological interpretation of the DMQ deposits and an initial discussion of local palaeoecology based on taxon representation.The current DMQ assemblage consists of the remains of minimally 147 individuals from 9 Orders and 14 Families of mammals. The carnivore assemblage described here is even more diverse than established in prior publications, including the identification of Megantereon whitei, Lycyaenops silberbergi, and first evidence for the occurrence of Dinofelis cf. barlowi and Dinofelis aff. piveteaui within a single South African site deposit. The cetartiodactyl assemblage is dominated by bovids, with the specimen composition unique in the high recovery of horn cores and dominance of Antidorcas recki remains. Other cetartiodactyl and perissodactyl taxa are represented by few specimens, as are Hystrix and Procavia; the latter somewhat surprisingly so given their common occurrence at penecontemporaneous deposits in the region. Equally unusual (particularly given the size of the sample) is the identification of single specimens of giraffoid, elephantid and aardvark (Orycteropus cf. afer) that are rarely recovered from regional site deposits. Despite the diversity within the DMQ macromammalian faunas, there are few habitat- or biochronologically-sensitive species that provide specific ecologic or age boundaries for the deposits. Recovered species can only support the non-specific, mixed open-to-closed palaeohabitats around Drimolen that have been reconstructed for the other penecontemporaneous South African palaeokarst deposits. The identified Equus quagga ssp. specimens recovered from the floor of the current excavation (∾−4.5–5 m below datum) suggests that most, if not all the DMQ specimens, were deposited after 2.33 Ma. Simultaneously, the carnivore specimens (D. cf. barlowi, L. silberbergi) suggest earlier Pleistocene (pre- 2.0–1.8 Ma) to maximally 1.6 Ma deposition (D. aff. piveteaui) for most of the DMQ fossil assemblage.

Introduction

The Drimolen fossil site was discovered in 1992 during survey of the region for a sinkhole by Dr. AW Keyser (then with the Geological Survey of South Africa; since 1993 known as the Council for Geoscience, one of the National Science Councils of South Africa) (Figs. 1 and 2). During his third site visit a hominin tooth was found from one of the ‘Main Pinnacles,’ which led to ad-hoc calcified sediment collection and mechanical preparation of fossil specimens from 1992 to mid-1994. In October 1994, a five-by-five metre alphanumeric grid was erected in the Main Quarry along with three fixed theodolite stations; shortly thereafter excavations into the decalcified material began that led to the recovery of the DNH 7 and 8 Paranthropus robustus specimens from the ‘Eurydice Pinnacle’ (Keyser et al., 2000). From 1995 until 2009, the site was excavated for 10 months out of the year, during which time changes to the grid system (e.g., numeric) and calcified sediment extraction methods (e.g., feather-and-wedge vs. drilling) were introduced. Although greatly reduced excavation seasons (∼6 weeks per year) have been employed from 2010 to the present, this nearly 25-year history of sampling the Main Quarry deposits has led to the recovery of a substantial assemblage of Paranthropus robustus and Homo sp. remains, as well as non-hominin primate and other faunal remains (Keyser et al., 2000; Gommery, Senut & Keyser, 2002; Lockwood et al., 2007; O’Regan & Menter, 2009; Moggi-Cecchi et al., 2010; Gallagher & Menter, 2011).

These excavations into the Main Quarry, and progressive exposure of the Drimolen Palaeocave System, has established that the site consists of a single large cavern in-filled with three main types of sedimentary deposits (speleothem, breccia, siltstone/sandstone) that often grade from one to the other laterally across the exposures (Fig. 2; Keyser et al., 2000). The base of the sequence consisted of a thick flowstone speleothem deposit that was extensively mined at the turn of the 20th century. This formed an undulating basal floor onto which all the other deposits and fossil were deposited. This basal speleothem consists of clast-poor layers of flowstone with a sharp contact to the overlying fossil-bearing sediments. As such, it formed a significant time before an entrance formed allowing the clastic deposits to enter the cave.

Figure 1 Aerial photograph and map of the Drimolen Palaeocave System relative to other penecontemporaneous South African fossil sites.

Figure 2 Schematic plan of the Drimolen Main Quarry excavation area and deposits.

In the centre of the Main Quarry a thick clast-supported, fossil-bearing breccia (referred to by Keyser et al. (2000) as the ‘Blocky Breccia’) occurred that represents the opening and subsequent collapse of a vertical entrance to the cave forming a central debris cone. This is best preserved against the western wall of the cavern where the deposits remain indurated (Fig. 2). The eastern part of this debris cone has suffered from collapse due to undercutting by lime miners removing the basal speleothem; as a result, the very eastern part of the Main Quarry was filled with mining debris and removal has accelerated the decalcification of these deposits.

During phases of flooding, the fine-grained fraction within the blocky breccia was then winnowed to the edges of the cavern to form a series of interstratified siltstone and sandstone deposits (referred to by Keyser et al. (2000) as ‘Cave Siltstone’). As a result, these two sedimentological deposits are contemporary and the clast-supported breccia of the central debris cone grades through a matrix-supported breccia and into the laminated siltstone and sandstone deposits towards the edge of the cavern. This exact sequence of formation can be seen at the nearby Wonder Cave active karstic system. The oldest siltstone and sandstone deposits in-filled the southern area of the Main Quarry. As the debris cone began to grow upwards and expand laterally it became less steep and matrix supported breccia became more prevalent. It is possible that at this time the original vertical shaft entrance, that would have initially restricted the access of certain animals to the cave, may have become so in-filled with sediments that entry became more accessible via a shallow talus cone into the fine-clastic sediment floors at the edge of the cavern. The southern area of the Main Quarry had become completely in-filled during this early phase due to the presence of a stepped cave roof, leading to concentrated deposition of the fine-grained siltstone and sandstone in the northern area of the cavern. It is notable that macrofaunal remains have not been recovered from the oldest southern siltstone and sandstone deposits, but are instead recovered either from the northern siltstone and sandstone deposits or from the central talus cone, suggesting that they could have been washed into the cave or that animals inhabiting the cavern stayed in the daylight zone around the entrance. Despite this there is little to suggest that the two periods are separated by any significant time interval, other than a series of thin (<2 cm) flowstones forming during the deposition of the northern siltstone and sandstone deposits.

Despite the decades of excavation into the Drimolen Palaeocave System, to date only a basic listing of macromammlian taxa recovered from the first years of Main Quarry deposits (Keyser et al., 2000) and a more comprehensive analysis of the Order Carnivora (O’Regan & Menter, 2009) have been previously published. This analysis provides the first primary description of the entire macromammalian (e.g., all Orders other than Macroscelidea, Insectivora, and Rodentia [exc. Family Hystricidae]) faunal assemblage excavated from the Drimolen Main Quarry since palaeontological sampling began in the early 1990s. We only present data on the fossil specimens attributable to at least Order as the taxonomically non-diagnostic remains from the deposits form the basis for a separate study reconstructing the taphonomy of the Main Quarry deposits.

Materials and Methods

The Drimolen Main Quarry (DMQ) fossil assemblage is curated in the Evolutionary Studies Institute at the University of the Witwatersrand and consists of 1,380 specimens derived from excavation of decalcified sediments and processed calcified sediment blocks (Keyser et al., 2000). Since the start of the excavations, the Drimolen Main Quarry decalcified material has been excavated in one-by-one meter (or smaller) squares in 10 cm increments and all material has been sieved through three different mesh sizes (the smallest being ∼1 mm). The sieving regimen, though, has undergone some changes over the years with wet sieving of the smallest mesh size from 1997 along with all these “fines” being sorted at a sorting table and not just quickly “over the sieve.” Since 2014, we have also wet sieved the medium fraction via a water pump and all the “fines” are now sorted in the lab over the following year. Prior to 2003, the only provenience for the decalcified material was by its grid square and 10 cm level boundaries. Since 2003, a total station theodolite has been permanently used at Drimolen allowing identifiable specimens or fragments larger than 2 cm to be directly piece-plotted in reference to a three-dimensional model of the deposits created via a Trimble laser scanner in 2012 (via the University of Cape Town Zamani Project) and again in 2015 with a Leica Nova MS50 multistation (from La Trobe University).

All calcified breccia blocks have been removed from the Main Pinnacles through standard feather-and-wedge techniques. Most of the calcified material was removed in the 1990s with no drilling of breccia after this period. Prior to 1997, this calcified material was either surveyed with a theodolite (without electronic distance meter) or at the very least the Pinnacle number or plane table location was recorded on the block. Since 1997, all calcified blocks were only sampled when a total station was available and were surveyed with at least six survey points. Breccia blocks within a decalcified excavation square were also recorded in this manner. The only other sampling of calcified material has been removing blocks that are decalcifying from the ‘Eurydice Pinnacle.’ This material is prepared mechanically until close to the bone and then the reduced block is prepared via acetic acid processing.

For this analysis we only considered specimens preserving sufficient morphology to be identified at least to Order level and are not presenting data on indeterminate mammalian elements or fragments. We have also not undertaken an analysis of the 554 primate craniodental and postcranial specimens, as this collection has recently been partially analysed (see Nieuwoudt, 2014). We have also not duplicated the primary description of the Main Quarry carnivores of O’Regan & Menter (2009), but we have re-evaluated the previously published Dinofelis and Chasmaporthetes specimens because of their bearing on biochronological interpretations and present data on carnivore specimens catalogued since publication of that study.

The taxonomic and/or element attributions were made in direct reference to the extant mammal skeletal materials of the Department of Vertebrates (Large Mammal, Small Mammal and Archaeozoology Sections) collections of the Ditsong National Museum of Natural History (Pretoria, South Africa), previously described fossil specimens in the Plio-Pleistocene Section of the Ditsong Museum and Evolutionary Studies Institute of the University of the Witwatersrand (Johannesburg, South Africa). All measurements of specimens reported here were taken using Mitutoyo 150 mm calipers with a direct digital input, including dental (MD, mesisodistal; BL, buccolingual; taken at the level of occlusion unless otherwise noted) and horn core metrics (AP, anteroposterior; ML, mediolateral; taken at the level of the pedicle unless otherwise noted). Three-dimensional (3D) surface scans were captured with an Artec Spider (Artec Group, Luxembourg) to facilitate morphological comparisons of some of the specimens following methods described in Adams et al. (2015). In some cases, external specimens were examined using a Dino-Lite Edge AM4815ZTZ microscope (AnMo Electronics Corp.).

In order to facilitate discussion and integrate the DMQ assemblage within the broader South African late Pliocene and Pleistocene fossil record, we draw from the extensive published literature on the well-described penecontemporaneous South African fossil deposit faunal assemblages (e.g., Bolt’s Farm, Gondolin, Haasgat, Kromdraai, Swartkrans, Sterkfontein). To facilitate comparisons of the proportion (based on the minimum number of individuals, MNI) of mammals in the DMQ deposits relative to other South African assemblages we have also calculated the McIntosh evenness statistic (McIntosh, 1967). We would note, however, that such direct comparisons of faunal assemblages assume that the taphonomic histories of these deposits (from predepositional processes and time-averaging to excavation/sampling methodologies) have not introduced substantial biases in the faunal representation or abundance; an assumption that may not always be warranted as each deposit has an idiosyncratic depositional history (see discussions in Brain, 1980; Reed, 1996; Pickering, 1999; Adams, 2006). As a comprehensive taphonomic analysis of the DMQ assemblage is still ongoing, we will not address the role taphonomic processes had in shaping the faunal presence/absence/abundance with the Main Quarry deposits at this time. We also largely do not consider the more recently described faunal assemblages (e.g., Motsetse (Berger & Lacruz, 2003), Hoogland (Adams et al., 2010), Malapa (Dirks et al., 2010; Kuhn et al., 2012)) as the faunal data from these localities is based on extremely small sample sizes and generally limited to simple listings without primary descriptions or supporting data.

Results

The current listing of identified non-hominin specimens (number of individual specimens, NISP; MNI) from the Drimolen Main Quarry is provided in Table 1, including primate specimens described in Nieuwoudt (2014) and carnivore specimens described in O’Regan & Menter (2009).

Table 1 List of macromammalian species from the Drimolen Main Quarry deposits.

Taxon	NISP	MNI	
Order primatesa			
Family Cercopithecidae			
Subfamily Colobinae			
Cercopithecoides williamsi	22	8	
Tribe Papionini			
Papio hamadryas robinsoni	260	45	
Cercopithecidae indet.	272	4	
Order Carnivorab			
Family Canidae			
Vulpes chama	1	1	
cf. Vulpes chama	1	–	
Canidae indet.	5	1	
Family Felidae			
Subfamily Machairodontinae			
Dinofelis aff. piveteaui	17	2	
?Dinofelis sp.	1	–	
Dinofelis cf. barlowi*	1	1	
Megantereon whitei*	2	1	
Machairodontinae indet.	1	1	
Subfamily Pantherinae			
Panthera pardus	5	1	
cf. Panthera pardus	3	1	
Panthera sp.	1	–	
Subfamily Felinae			
cf. Caracal caracal	1	1	
Felis silvestris lybica	3	1	
cf. Felis silvestris lybica	4	2	
Felidae indet.*	31	–	
Family Herpestidae			
aff. Suricata suricatta	1	1	
cf. Cynictis penicillata	3	1	
Family Hyaenidae			
Chasmaporthetes nitidula*	1	1	
Lycyaenops silberbergi*	1	1	
Hyaenidae indet.*	7	2	
Carnivora indet.	15	–	
Order Cetartiodactyla			
Family Bovidae			
Tribe Alcelaphini			
Connochaetes sp.	2	2	
Damaliscus sp.	2	2	
Megalotragus sp.	4	2	
Indeterminate (Class II/III)	34	9	
Indeterminate (Class III)	11	4	
Tribe Antilopini			
Antidorcas recki	25	16	
Antidorcas cf. recki	26	2	
Raphicerus sp.	1	1	
Tribe Oreotragini			
Oreotragus sp.	12	9	
Tribe Reduncini			
Redunca cf. fulvorufula	3	3	
Tribe Tragelaphini			
Tragelaphus sp.	11	6	
Indeterminate	2	–	
Bovidae indet.	538	–	
Family Giraffidae			
Giraffidae indet.	1	1	
Family Suidae			
Suidae indet.	1	1	
Order Perissodactyla			
Family Equidae			
Equus quagga ssp.	3	1	
Order Proboscidea			
Family Elephantidae			
Elephantidae indet.	1	1	
Order Hyracoidea			
Family Procaviidae			
Procavia sp.	5	3	
Order Lagomorpha			
Family Leporidae			
Pronolagus sp.	1	1	
Leporidae indet.	36	5	
Order Rodentia			
Family Hystricidae			
Hystrix sp.	1	1	
Order Tubulidentata			
Family Orycteropodidae			
Orycteropus cf. afer	1	1	
Total	1,380	147	
Notes.

a Taxonomic attributions, specimen counts, and MNI values as reported by Nieuwoudt (2014).

b Taxonomic attributions, specimens counts, and MNI values as reported by O’Regan & Menter (2009) except where amended here (amended taxa and/or counts marked with an asterisk).

Systematic Palaeontology

Order CARNIVORA Bowditch, 1821	
Family FELIDAE Batsch, 1788	
Subfamily MACHAIRODONTINAE Gill, 1872	
Genus DINOFELIS Zdansky, 1924	
Type species Dinofelis abeli Zdansky, 1924	
Dinofelis cf. barlowi Broom, 1937	

Referred specimens. DN 2791, right partial maxillary canine.

Description. This single partial canine is the first specimen from the site to be considered comparable to Dinofelis barlowi (Fig. 3A). The canine exhibits greater transverse crown compression than present in pantherines like extant and contemporaneous Panthera pardus Linnaeus, 1758 canines from Swartkrans Member 1 (e.g., SK 349 [0.77], SK 354 [0.75]) (Table 3; Fig. 4). The size and transverse crown compression of DN 2791 is consistent with the machairodonts Dinofelis piveteaui Ewer, 1955a KA 61 type specimen (0.59; BL: 12.1 mm/MD: 20.5 mm) and Dinofelis barlowi BF-55 22 specimen (0.59; BL: 14.5 mm/MD: 24.5 mm). In contrast to the canines of the KA 61 D. piveteaui specimen, the DN 2791 specimen is both more curved from root to crown, more oval in cross-section, and lacks the well-developed distal carina. The DN 2791 specimen is, however, identical to BF-55 22 in root-to-crown curvature, the ovoid cross-sectional shape, and the development of the mesial and distal carinae. Based on these comparisons we provisionally attribute the canine to D. cf. barlowi. We discuss the significance of this novel specimen relative to the previously published DMQ Dinofelis aff. piveteaui craniodental and postcranial specimens (O’Regan & Menter, 2009) below.

Figure 3 Order Carnivora specimens attributed to the Family Felidae from the Drimolen Main Quarry.

(A) DN 2791 Dinofelis cf. barlowi right maxillary canine, buccal (left) and distal (right) views. (B) DN 976 Megantereon whitei left P4, buccal (left) and lingual (right) views. (C) DN 3254 Megantereon whitei left mandibular corpus, lateral (left) and anterior oblique (right) view of surface scan to highlight sharp diastema margin and mandibular flange. Scale bars equal 1 cm.

Figure 4 Bivariate plot of select Felidae maxillary canine MD length on BL width (mm), including DN 2791 Dinofelis cf. barlowi.

Metrics and regression equations presented in Table 3.

Genus MEGANTEREON Croizet and Jobert 1828	
Type species Megantereon cultridens Cuvier, 1824	
Megantereon whitei Broom, 1937	

Referred specimens. DN 976, left partial P4; DN 3254, left mandibular symphysis and corpus with partial canine.

Description. Two specimens represent the machairodont Megantereon whitei from the Drimolen Main Quarry deposits. The DN 976 left P4, while broken and reglued with slight distortion on the lingual aspect, preserves part of the protocone, paracone and metastyle (Fig. 3B; Table 2). The protocone is extremely small and grades smoothly into the upright paracone. There is a deep notch separating the paracone and bulbous metacone, and a distinct notch on the metastyle. This morphology is shared with the KA 64 Megantereon whitei P4 to the exclusion of other machairodonts (e.g., Dinofelis piveteaui, Dinofelis barlowi), similarly-sized extant felids (e.g., Panthera pardus, Acinonyx jubatus Schreber, 1775), or the indeterminate Main Quarry felids (DN 530, 4300, 5498) described below.

The DN 3254 left partial mandible preserves part of the symphysis and alveolus for the i1-i3, the canine alveolus with part of the canine root, and the diastema (Fig. 3C). Although this specimen preserves the anterior portion that does not overlap other South African M. whitei mandibles (e.g., KA 64, TM 856, STS 1588; Ewer, 1955a; Turner, 1987a), several morphological features align the specimen with M. whitei; the most diagnostic of these features is the presence of a mandibular flange on the anterolateral border of the corpus adjacent to the canine alveolus. The root of the canine indicates the tooth was small (length: 12.5 mm, breadth: 7.1 mm) and consistent with other African and Eurasian Megantereon specimens (Palmqvist et al., 2007). Finally, the diastema is sharply margined and elongated given the absence of any premolar alveolus despite the preserved length of the corpus.

Felidae gen. et sp. indet.	

Referred specimens. DN 530, right P4; DN 558, left proximal ulna; DN 2236, terminal phalanx; DN 2701, left partial mandible; DN 2937, indet. maxillary incisor; DN 3291, right calcaneus; DN 4300, right P4 (probable antimere to DN 5498); DN 4354, two associated indet. mandibular incisors; DN 4590, right partial astragalus; DN 5498, left P4 (probable antimere to DN 4300).

Description. A small collection of carnivore craniodental and postcranial specimens are attributable to the Family Felidae but are not considered sufficiently diagnostic to attribute to genus or species. The DN 558 left proximal ulna preserves part of the olecranon process and articular surface and is derived from a large, Dinofelis-sized felid but is too damaged to confidently attribute. Also likely derived from a larger felid is the DN 3291 partial calcaneus and DN 4590 partial astragalus, with the latter similar to the previously described DN 2149b D. aff. piveteaui astragalus (O’Regan & Menter, 2009: 338). In contrast, the DN 2236 phalanx is from a smaller felid species that appears slightly larger than extant caracal (Caracal caracal Schreber, 1776) and preserves a robust volar surface. The DN 2937 maxillary incisor is tall, pointed, and distinctly tricusped, preserving parts of two discrete lingual accessory cusps and is consistent with derived machairodont incisors (e.g., BF-55 23 D. barlowi, KA 61 D. piveteaui, KA 64 Megantereon whitei; Biknevicius, Van Valkenburgh & Walker, 1996; Christiansen & Adolfssen, 2007); the similarly-shaped DN 4354 mandibular incisors are also likely derived from a machairodont felid, perhaps the same individual, given their shared size and distinct morphology.

Table 2 Measurements (in mm) of Drimolen Main Quarry Felidae gen. et sp. indet and comparative maxillary fourth premolars body.

Specimen	Deposit	MD	BL	Wpc	PaL	MtL	
Felidae gen. et sp. indet.	
DN 4300	Drimolen Main Quarry	29.8	11.6	9.4	11.8	12.3	
DN 5498	Drimolen Main Quarry	30.2*	11.6*	9.3	11.1*	11.9	
Megantereon whitei	
DN 976	Drimolen Main Quarry	23.4*	11.2*	9.6	10.0	10.6	
KA 64	Kromdraai A	29.7	11.1*	9.7*	9.6	10.8	
Dinofelis piveteaui and aff. piveteaui	
DN 1012	Drimolen Main Quarry	36.5	13.9		13.5	14.3	
KA 61	Kromdraai A	41.0	13.0			17.8	
MT 1986a	Motsetse	38.8	12.8			16.5	
Dinofelis barlowi	
BF 55-22	Bolt’s Farm Pit 23	37.0*	16.5			14.0	
SF 5855a	Sterkfontein Member 4	35.2	15.5		13.0	13.7*	
Notes.

* Minimum value given damage to measured region.

a Measurements of Motsetse and Sterkfontein Member 4 specimens derived from Lacruz, Turner & Berger (2006).

Table 3 Regression equations and metrics for the Felidae canine shape groups in Fig. 4.

Maxillary canine MD length and BL width (in mm), along with the canine compression ratio (BL/MD).

Canine morphology	Regression equation	R2	
Conical-toothed	y = 0.7533x + 1.2559	0.9567	
False Saber-toothed	y = 0.7119x − 2.0968	0.9096	
Saber-toothed	y = 0.495x − 0.01	0.8754	
	MD length (mm)	BL width (mm)	Compression ratio	
Conical-toothed Felids			0.83	
Puma concolor(n = 28)a	12.9	11.7	0.91	
Neofelis nebulosa(n = 18)a	11.5	8.2	0.72	
Panthera uncia(n = 12)a	11.9	11.1	0.93	
Panthera tigris(n = 28)a	24.3	19.7	0.81	
Panthera onca(n = 20)a	19.1	15.9	0.83	
Panthera pardus(n = 26)a	14.7	12.4	0.85	
Panthera leo(n = 17)a	23.3	18.3	0.78	
False Saber-toothed Felids			0.61	
Dinofelis petteri(n = 2)b	21.0	12.7	0.60	
Dinofelis aronokib	20.4	12.8	0.63	
Dinofelis barlowi(n = 4)b	24.1	14.6	0.61	
Dinofelis piveteaui	20.5	12.2	0.59	
Dinofelis cf. diastemata(n = 2)b	20.2	11.9	0.59	
Dinofelis cristata(n = 2)b	25.3	16.9	0.67	
Dinofelis palaeooncab	18.6	11.7	0.63	
Dinofelis sp.b	24.8	15.1	0.61	
DN 2791	23.2	14.1	0.61	
Saber-toothed Felids			0.51	
Homotherium seruma	34.6	15.8	0.46	
Megantereon cultridens(n = 27)a,c	22.6	11.6	0.51	
Megantereon whitei(n = 8)c	24.1	12.3	0.51	
Smilodon fatalisc	44.2	24.7	0.56	
Smilodon populatorc	42.8	19.0	0.44	
Notes.

a Christiansen (2007).

b Werdelin & Lewis (2001).

c Palmqvist et al. (2007).

Four of the indeterminate felid craniodental specimens may ultimately be attributable below the level of Family with additional analysis beyond this primary description. The DN 2701 left mandible preserves the posterior portions of the corpus, including gonion and the condylar and coronoid processes. The masseteric fossa is deep and exhibits robust scarring both dorsal and ventral to the margins. The specimen is much smaller than D. barlowi (BF-55 23) and D. piveteaui (KA 62, KA 63) mandibular remains, but similar in size to both M. whitei (KA 64, STS 1588) and P. pardus (extant comparative and SK 349) mandibles. However, in contrast with extant P. pardus mandibles (and SK 349) the DN 2701 coronoid process is shorter, narrower anteroposteriorly, and less robust. In addition, the DN 2701 gonion exhibits less lateral flare and the condyle is more robust and angled relative to the axis of the corpus. The DN 2701 coronoid process is also smaller and less robust than that of the KB 5224b M. whitei mandible from Kromdraai B (Turner, 1987a), but shows a similar condylar morphology to both the KB 5224b and KA 64 specimens. Unfortunately, this region is poorly preserved or absent in other South African Megantereon mandibles to assess potential variability in coronoid process morphology that may eventually provide for a more confident attribution of the DN 2701 specimen.

The DN 530, 4300 and DN 5498 maxillary fourth premolars all preserve similar cusp morphologies (DN 4300 and DN 5498 are probable antimeres based on shared morphology and spatial position in the deposits) and are likely derived from the same felid species (Fig. 5A). The DN 530 specimen exhibits damage to the crown elements, and the DN 5498 P4 was broken in three places and reglued; however, the latter retains some of the maxillary alveolar bone including the alveolus for the M1, which indicates a robust, ovoid and possibly double-rooted M1 in life. All of these specimens exhibit weakly-developed protocones that grade smoothly into the paracone, and a deep cleft dividing the paracone and metastyle. They are all larger than extant Acinonyx jubatus and Panthera pardus (comparative measurements in Turner, 1984; Table 2) and also differ in protocone size (e.g., more developed than in A. jubatus, less separated from the paracone and more distally positioned than in P. pardus). None of these specimens can be accommodated into Dinofelis based on their size and morphology of the parastyle and metastyle (Table 2). The closest metric and morphological match for these Main Quarry specimens is the KA 64 Megantereon whitei P4; however, the DN specimens exhibit a parastyle more distinct from the paracone, a more rudimentary ectostyle, no buccolingual pinching of the metastyle, and a flared posterior lip of the metacone (Fig. 5A). Lacking a strong morphological match for these specimens, we have deferred a more specific taxonomic attribution of the specimens pending further analysis.

Figure 5 Order Carnivora specimens attributed to the Family Felidae and Hyaenidae from the Drimolen Main Quarry.

(A) DN 4300 Felidae gen. et sp. indet. right P4, buccal view. (B) DN 974 Lycyaenops silberbergi right P3, occlusal (top), lingual (left) and buccal (right) views. Scale bars equal 1 cm.

Family HYAENIDAE Gray, 1821	
Genus LYCYAENOPS Kretzoi, 1938	
Type species Lycyaenops rhomboideae Kretzoi, 1938	
Lycyaenops silberbergi Broom in Broom and Schepers, 1946	

Referred specimens. DN 974, right P3.

Description. A primary description of the DN 974 specimen was provided in O’Regan & Menter (2009), although we differ in treating the DN 974 specimen as a right P3 rather than a right P2 based on size and morphology (Fig. 5B; Table 4). In the original description, O’Regan & Menter (2009: 343–344) allocated DN 974, and the DN 404 partial cranium, to Chasmaporthetes nitidula Hay, 1921 based on dental metrics and interpretation of mesial accessory cusp and P4 morphology relative to the Swartkrans Member 1 C. nitidula specimens (specifically SK 305, 306, and 307) and previously published descriptions by Broom (1948) and Ewer (1955b).

In contrast to the prior taxonomic attribution, the preserved morphology of the DN 974 specimen is most consistent with the type specimen and other African specimens of Lycyaenops silberbergi. The DN 974 right third premolar preserves a tall paracone and a large distal accessory cusp, but no mesial accessory cusp and only a modest mesial cingulum. The crown is rectangular in outline, has a distally-shifted lingual shelf well-separated from the paracone, and the paracone is separated from the distal accessory cusp by a distinct ‘waist,’ particularly well-developed on the labial aspect. This morphology is shared with the L. silberbergi P3 type (STS 130) and other specimens from Sterkfontein Member 4 (STS 127, 130, 135; SF 369/372, 383/373, 408) and the Lycyaenops cf. L. silberbergi from Laetoli (NHM AS 7.VI.35, LAET 75-494) (Turner, 1986; Turner, 1990; Werdelin & Dehghani, 2011). It differs from the C. nitidula P3s from Swartkrans Member 1 (SK 305, 306, 309, 312, 313) and Member 3 (SKX 29205) in the development of the strong waist on the labial aspect and generally in mesial accessory cusp morphology (Ewer, 1955b; Brain, 1980; Turner, 1993). We echo prior publications that there is some variability in mesial accessory cusp expression across C. nitidula maxillary premolars (e.g., Brain, 1980: 234 ‘type A, primitive’ and ‘type B, advanced’; Werdelin & Peigné, 2010); however, only SK 305 and 306 preserve a more reduced mesial accessory cusp (the former is damaged in this region) and both have more developed mesial cusps than that seen on DN 974. As noted by O’Regan & Menter (2009) the DN 404 dentition is damaged, but the P3 has a distinctly oval profile and smooth labial contour; the mesial and lingual aspects of the crown are somewhat crushed and offset, prohibiting confident morphological assessment but likely exhibited a modest mesial accessory cusp (and is closely resembling the ‘type A, primitive’ Swartkrans Member 1 SK 306 partial maxilla).

Hyaenidae gen. et sp. indet.	

Referred specimens. DN 2864, partial right scapula; DN 2973, partial right P4; DN 3281, partial right P2.

Description. In addition to the four indeterminate hyaenid craniodental and postcranial specimens described by O’Regan & Menter (2009), three further elements can be attributed to the family. The DN 2864 scapula preserves a very large infraglenoid tubercle relative to the size of the preserved glenoid fossa and is derived from hyaenid smaller than extant Parahyaena brunnea Thunberg, 1820. The DN 2973 right P4 preserves the complete protocone and lingual aspect of the anterior accessory cusp and paracone with little occlusal wear. There is some buccolingual swelling on the paracone that is shared with extant P. brunnea P4s and the SK 327 P. brunnea P4 from Swartkrans Member 1 (although smaller than the latter), and there is no evidence for buccolingual compression or a ridge leading to the trigon basin as in Crocuta crocuta. This specimen, along with the DN 2321 P4 fragment described by O’Regan & Menter (2009: 344), support the occurrence of a hyaenid individual distinct from Lycyaenops and Chasmaporthetes in the Main Quarry deposits. In contrast, the DN 3281 right P2 preserves the anterior margin of the crown and half of the anterior root that lacks the cingulum distinctive for P. brunnea dentition. The specimen preserves strong labial ridging and a flattened lingual aspect, and is too buccolingually expanded to represent Chasmaporthetes or Lycyaenops. The closest extant morphological match is Crocuta crocuta; minimally suggesting an additional hyaenid distinct from L. silberbergi, C. nitidula, and the DN 2973/2321 individuals.

Order CETARTIODACTYLA Montgelard et al., 1997	
Family BOVIDAE Gray, 1821	
Tribe ALCELAPHINI de Rochebrune, 1883	
Genus CONNOCHAETES Lichtenstein, 1814	
Type species Connochaetes gnou Zimmermann, 1780	
Connochaetes sp.	

Referred specimens. DN 704, left maxillary third molar; DN 1111a, right maxillary second molar.

Description. Only two of the identifiable alcelaphin specimens represent the genus Connochaetes in the Main Quarry sample. The DN 704 third molar, derived from decalcified sediments, is a complete crown (ML: 29.8 mm, BL: 19.0 mm) that exhibits moderate occlusal wear. The DN 1111a molar is also a complete crown (ML: 25.5 mm) with moderate occlusal wear but is set within a partially mechanically processed aggregation of indeterminate postcranial fragments. As isolated teeth the specimens cannot be confidently attributed below the generic level. Both specimens are consistent in cusp morphology, central cavity complexity and overall size with extant comparatives and fossil Connochaetes sp. specimens from Swartkrans Members 1 Hanging Remnant (e.g., SK 2482, 3008, 3102, 14120x) (linear metrics in Vrba, 1976).

Genus DAMALISCUS Sclater and Thomas, 1894	
Type species Damaliscus dorcas Pallas, 1766	
Damaliscus sp.	

Referred specimens. DN 2790, right horn core; DN 4778, right horn core.

Description. Two partial right horn cores, representing two adult individuals, are attributed here to the genus Damaliscus. The most complete of the cores, DN 2790, preserves the pedicle (infiltrated by an extensive sinus) to approximately ¾ of the body of the core, which exhibits a small amount of torsion approaching the broken margin (Fig. 6A). The base of DN 2790 core is mediolaterally compressed (AP: 35.3 mm, ML: 26.3 mm) and there is a deep medial groove on the body. The DN 4778 specimen preserves the body of the core in two articulating pieces, with a third associated fragment lacking a clear contact point. While mechanical preparation damage to DN 4778 prohibits confident metric comparison it exhibits analogous mediolateral compression to the DN 2790 specimen.

Figure 6 Order Cetartiodactyla specimens attributed to the Family Bovidae from the Drimolen Main Quarry.

(A) DN 2790, Damaliscus sp. right horn core, medial view. (B) DN 748a and 748b, Megalotragus sp. right M2 and M3, occlusal view. (C) DN 1015, Alcelaphini gen. et sp. indet. left mandible, occlusal view. (D) DN 1013 Alcelaphini gen. et sp. indet. right horn core, anterior (probable) view. Scale bars equal 1 cm.

Both the DN cores exhibit similar size, curvature and mediolateral compression to extant Damaliscus dorcas comparative specimens and the SK 14206 specimen attributed to Damaliscus cf. dorcas by Vrba (1976) (ML: ∼24.0 mm). Both Main Quarry specimens contrast with previously described Damaliscus niro Hopwood, 1936 horn cores (including the Swartkrans Member 2 specimen SK 2862) in size (e.g., SK 2862, AP: ∼44.0 mm, ML: ∼36.0 mm; Vrba, 1976; published metrics from Olduvai and Cornelia in Gentry & Gentry, 1978), the onset of torsion and the absence of transverse ridging (see also Cooke, 1974; Gentry, 2010). Although these two specimens are morphologically close to D. dorcas, we agree with Gentry (2010: 786) that a comprehensive revision of smaller Damaliscus from Pleistocene localities is critical before attributing specimens like the Main Quarry horn cores to a species within the genus.

Genus MEGALOTRAGUS van Hoepen, 1932	
Type species Megalotragus priscus Broom, 1909	
Megalotragus sp.	

Referred specimens. DN 748a, right M2; DN 748b, right M3; DN 856, left M2 (?); DN 4807, left M2.

Description. Four isolated alcelaphin maxillary molars exhibit dental dimensions that exceed extant Connochaetes comparatives with analogous occlusal wear and are consistent with fossil specimens attributed to the extinct genus Megalotragus. The two associated right molars (DN 748a, b) preserve complete crowns (DN 748a ML: 27.9 mm, BL: 19.2 mm; DN 748b ML: 36.0 mm, BL: 19.4 mm) and moderate-heavy occlusal wear (Fig. 6B). Both DN 4807 and DN 856 are left M2s (with the DN 856 position in the toothrow only tentative) with moderate occlusal wear. While DN 4807 is missing the lingual enamel surface and is somewhat distorted, DN 856 preserves a nearly complete probable M2 crown (ML: 31.6 mm, BL: 24.4 mm) (with regluing and minor enamel damage). All four of these specimens are directly comparable in size and cusp morphology to Megalotragus maxillary remains from Swartkrans Member 1 Hanging Remnant (e.g., SK 2245, 2432, 3031) and Member 2 (e.g., SK 14120, 14218) (linear metrics in Vrba, 1976).

Alcelaphini gen. et sp. indet.	

Referred specimens. DN 3, left deciduous P3; DN 41, left maxillary molar; DN 42, left maxillary molar (associated with DN 41); DN 82, left M3; DN 90, left deciduous P3; DN 255, horn core fragment; DN 259, right m1; DN 309, left M1 or M2; DN 446, right m1; DN 475, right deciduous p4; DN 529, right mandible with deciduous p3-p4; DN 719, right P3; DN 722, right m1 or m2; DN 831, right (?) horn core fragment; DN 837, mandible fragment; DN 863, cranial fragment; DN 878, horn core fragment; DN 1001, left (?) horn core fragment; DN 1006, right deciduous p3; DN 1007, left m3; DN 1013, right horn core fragment; DN 1015, left mandible with deciduous p2-p4; DN 1021, right P3; DN 1025, deciduous P4; DN 1026, left m1 or m2; DN 1043, horn core fragment; DN 1061, left M2; DN 1065, right deciduous p3; DN 1099, left (?) horn core fragment; DN 1143, left mandible with deciduous p2-p4; DN 1156, indet. side M2 or M3; DN 2053, horn core pedicle fragment; DN 2157, right M1 or M2; DN 2161, horn core tip fragment; DN 2168, tooth fragment; DN 2854, right m3; DN 2992, left mandibular molar fragment; DN 4321, horn core fragment; DN 4466, right deciduous p4; DN 4514, right deciduous p4; DN 4526, left m3; DN 4647, molar fragment; DN 4653, left mandible with p3-p4; DN 4779, right M3; DN 4780, left m3.

Description. Because of either incomplete preservation and/or the metric and morphological overlap in the isolated dentition of extant and extinct alcelaphin genera and species, a total of 45 Main Quarry specimens could not be confidently attributed below the level of the Tribe. Within this sample, 11 specimens (DN 259, 529, 722, 863, 878, 1099, 1156, 2157, 2161, 2992, and 4653) are derived from minimally four larger size class III individuals (e.g., within the size range of extant Connochaetes or larger); all adults excepting DN 529, which is an extremely young individual with the deciduous premolars just erupting. The remaining specimens are derived from minimally nine larger class II and smaller class III alcelaphin individuals (e.g., within the size range of extant Damaliscus and Alcelaphus), including at least seven immature individuals retaining their deciduous dentition (e.g., DN 1015; Fig. 6C).

At present we include in this group five partial horn cores that may ultimately be attributable to the generic or specific level; however, the paucity of horn cores from penecontemporanous South African fossil deposits (particularly relative to the eastern African record) limits our ability to confidently diagnose the specimens. The DN 4321 horn core exhibits a very large sinus within an expanded pedicle, but supporting a very compressed core body that somewhat resembles the morphology of extant Sigmoceras lichtensteinii Peters, 1849 (alt. Alcelaphus buselaphus lichtensteinii sensu Gentry, 2010; Kingdon & Hoffmann, 2013b). DN 1043 is a left, likely immature horn core (AP: 39.3 mm) that is derived from a smaller alcelaphin and somewhat resembles the immature cores of extant Damaliscus lunatus Burchell, 1823 as well as the SK 14008 indeterminate bovid horn core from the Member 1 Hanging Remnant.

The three remaining horn cores (DN 255, 381, and 1013) appear to exhibit the same morphology and are likely derived from the same alcelaphin species. The most complete of these, DN 1013, is a ∼10 cm portion of a right horn core pedicle and body (Fig. 6D). The base of the horn core lacks bossing as in extant Alcelaphus or Connochaetes, and the strong anticlockwise torsion eliminates attribution to extant Damaliscus as well as the Swartkrans Member 1 SK 3211b Numidocapra porrocornutus Vrba, 1971 specimen (sensu Gentry, 2010 after Vrba, 1997), Parmularius braini Vrba, 1977, or Damaliscus gentryi Vrba, 1977 as represented by specimens from Makapansgat Member 3. Amongst the previously described alcelaphin fossil horn cores, the closest comparative specimen is the SK 14183 Beatragus sp. Heller, 1912 from Swartkrans Member 2 (Vrba, 1976). When oriented relative to SK 14183, DN 1013 exhibits features previously used by Vrba (1976) when attributing the SK specimen to Beatragus: anteroposterior flattening of the core ∼3 cm above the base, anticlockwise torsion, a modest mesial keel, foramina along the anterolateral border, and slight swelling on the posterolateral aspect (which is compromised by surface damage) that may indicate the origin of a posterolateral keel. We note, however, that the DN 1013 specimen is smaller (AP: 37.8 mm, ML: 45.4 mm) than the SK 14183 Beatragus specimen (AP: 45.2 mm, ML: 51.2 mm) and the torsion is more exaggerated; potentially reflecting ontogenetic, demographic or phylogenetic differences.

Tribe ANTILOPINI Gray, 1821	
Genus ANTIDORCAS Sundevall, 1847	
Type species Antidorcas marsupialis Zimmermann, 1780	
Antidorcas recki Schwarz, 1932	

Referred specimens. DN 224, left adult male horn core; DN 879, left male horn core; DN 884, right immature male horn core; DN 890, right adult male horn core; DN 938, left adult female horn core; DN 990, right (?) immature male horn core; DN 995, left adult female horn core; DN 1014, right sub(?)adult female horn core; DN 1022, left adult male horn core; DN 1048, indet. side female horn core; DN 1055, right (?) immature male horn core; DN 1058, left sub(?)adult male horn core; DN 1060, right adult male horn core; DN 1068, left adult male horn core; DN 1071, left sub(?)adult male horn core; DN 2483a left adult female horn core; DN 2483b right adult female horn core; DN 2789, indet. side male horn core; DN 3033, indet. side female horn core; DN 3294, right immature male horn core; DN 4438, right adult female horn core; DN 4690, left immature male horn core (possible antimere of DN 884); DN 4698, indet. side female (?) horn core; DN 4777, right male horn core; DN 4796, left adult female horn core.

Description. At present, this collection of 25 Antidorcas recki horn core specimens from at least 16 individuals is the largest described sample for the species from the Cradle (and outside the Olduvai Bed deposits) (Gentry, 1966; Gentry & Gentry, 1978; Cooke, 1996). It is also the most demographically diverse from the South African deposits, with minimally eight males (three adult, two probable subadult, three immature) and eight females (seven adult, one probable subadult) recorded in the sample.

The morphology of the horn cores largely conform to the description of male and female A. recki specimens from Olduvai (Gentry, 1966; Gentry & Gentry, 1978) and Bolt’s Farm Pit 3 (Cooke, 1996) (and contrasts with extant Antidorcas marsupialis horn cores) in the vertical rise of the horn core from the sinus-filled pedicle, degree of mediolateral compression, strong posterior angulation within the short body, and a lack of lateral divergence or torsion (Figs. 7A and 7B; Table 5). The adult and immature male DN specimens contrast with those of Gazella vanhoepeni Wells and Cooke, 1956 from the Makapansgat Member 3 deposits (e.g., M 412, 415, 2224, 2717, 8245, 8384, 9006, 9026) in the less pronounced mediolateral compression, core body size, and the sharper posterowards curvature in the core (Fig. 7A). We do note, however, that none of the male DN horn cores exhibit the strong transverse ridging present on the Bolt’s Farm Pit 3 male cranium (UCMP 69521), which aligns the DN specimens with some of the noted A. recki variants from Olduvai and Peninj described by Gentry & Gentry (1978: 429). The substantial sample of female A. recki horn core specimens in the Main Quarry sample provides further evidence for the presence of strong sexual dimorphism in this skeletal feature within the species (Cooke, 1996). The DN female cores are substantially smaller, straighter, and exhibit a far more rounded cross-section than the male counterparts (Fig. 7B); consistent with the morphology of the Bolt’s Farm Pit 3 UCMP 80168 and 80169 specimens (Cooke, 1996; Table 5).

Figure 7 Order Cetartiodactyla specimens attributed to the Family Bovidae from the Drimolen Main Quarry.

(A) DN 1060, Antidorcas recki right male horn core, medial view (left); DN 995, Antidorcas recki left female horn core, medial view (right). (B) DN 438, Oreotragus sp. right mandible, occlusal view. (C) DN 120, Tragelaphus sp. right mandible, occlusal view. Scale bars equal 1 cm.

Table 4 Comparative measurements (mm) of African fossil Chasmaporthetes and Lycyaenops maxillary premolars.a

		P2	P3	P4	
Specimen	Deposit	MD	BL	MD	BL	MD	BL	
Chasmaporthetes nitidula								
DN 404	Drimolen Main Quarry			20.2*	11.6*	31.5	13.3	
SK 305	Swartkrans Member 1	15.8*	10.7	20.3*	12.9	32.7	15.3	
SK 306	Swartkrans Member 1			19.6	13.6	32.4*	14.8	
SK 307	Swartkrans Member 1					32.4	14.5	
SK 309	Swartkrans Member 1			21.1	14.6			
SK 310	Swartkrans Member 1	18.8	11.9					
SK 311	Swartkrans Member 1	18.4	12.5					
SK 312	Swartkrans Member 1			21.9	14.0		15.8*	
SK 313	Swartkrans Member 1			20.8	14.4			
SK 379	Swartkrans Member 1	19.5	12.0					
SKX 29205	Swartkrans Member 3			22.4	12.8*			
SKX 22992/72	Swartkrans Member 3					30.8*	15.8	
SF 363	Sterkfontein Member 4						15.4	
Lycyaenops silberbergi								
DN 974	Drimolen Main Quarry			20.4	13.5			
STS 130	Sterkfontein Member 2b			23.8	14.5*			
STS 135	Sterkfontein Member 4			19.6	11.9			
SF 383/373	Sterkfontein Member 4			22.6				
SF 369/373	Sterkfontein Member 4			22.8				
SF 408	Sterkfontein Member 4			20.3				
SF 463	Sterkfontein Member 4	17.1	11.0					
Lycyaenops cf. L. silberbergi								
NHM AS 7.VI.35	Laetoli (Laetolil Beds, Upper Unit)			17.7	12.5			
LAET 75-494	Laetoli (Unknown level)	19.1	11.0					
Notes.

* Minimum value given damage to measured region.

a All measurements reported are by the authors except those for Sterkfontein (SF) premolars reported by Turner (1987b) and Laetoli premolars reported by Werdelin & Dehghani (2011).

b The stratigraphic origin of the STS 120 L. silberbergi type specimen from Sterkfontein is uncertain but may be derived from Member 2 or 3 (Brain, 1980).

Antidorcas cf. recki	

Referred specimens. DN 140, right indet. maxillary molar; DN 175, right m1 or m2; DN 308, right m3; DN 401, left P4; DN 447, right m1; DN 989, right P3; DN 1002, right mandible with p4-m2 (associated with DN 2283); DN 1034, right m1 or m2; DN 1054, right M1 or M2; DN 1059, indet. side male (?) horn core; DN 1115, right immature horn core; DN 1116, right m3; DN 1135, left P4; DN 2175, indet. side m1 or m2; DN 2270, left mandible with p4 and m2; DN 2271, right P4; DN 2283, right m3; DN 2307, right immature horn core; DN 2328, left p4; DN 2554, left M2; DN 2614, right mandible with deciduous p2-p4 and m1-m2; DN 3009, left P3; DN 3287, right m2; DN 3346, left m1 or m2; DN 4185, right M2; DN 4290, right m3.

Description. In addition to the attributed Antidorcas recki horn cores, a collection of 26 Antidorcas craniodental specimens have also been recovered from the Main Quarry deposits. This sample of isolated teeth display similar crown morphology with other previously attributed Antidorcas recki dentition from Bolt’s Farm (see above; Cooke, 1996), Kromdraai A (e.g., KA 964B, 1002, 1111, 1093) and dissimilar to both extant Antidorcas marsupialis Zimmerman, 1780 and the hypsodont Antidorcas bondi Cooke and Wells, 1951 specimens from Swartkrans Member 2 (Vrba, 1976). Furthermore, most of these isolated specimens were recovered interspersed within the decalcified Main Quarry sediments alongside the diagnostic A. recki horn core specimens; and in the case of DN 3346 and 4185, were recovered from just adjacent to the DN 3033 and 3294 horn cores. We do, however, recognise and support the caution expressed by Gentry (2010) regarding the separation of remains of A. recki from early remains of A. bondi and Antidorcas australis Hendey and Hendey, 1968/A. marsupialis. In keeping with a more conservative approach to these fragmentary craniodental remains, we treat them as provisionally attributable to A. recki.

Genus RAPHICERUS C.H. Smith, 1827	
Type species Raphicerus campestris Thunberg, 1811	
Raphicerus sp.	

Referred specimen. DN 591, right p3.

Description. The single right mandibular third premolar is a complete crown (ML: 7.7 mm, BL: 3.1 mm) with only superficial occlusal wear. The tooth is very mesiodistally elongate, exhibiting the gracile cusp morphology and weak-development of distal elements (e.g., hypoconid, entoconid/entostylid barely developed and fused) that is similar to the equivalent tooth preserved in the KA 710 Raphicerus campestris mandible from Kromdraai A (Vrba, 1976) and extant R. campestris comparatives.

Tribe OREOTRAGINI Pilgrim, 1939	
Genus OREOTRAGUS A. Smith, 1834	
Type species Oreotragus oreotragus Zimmermann, 1783	
Oreotragus sp.	

Referred specimens. DN 290, left mandible with deciduous p3; DN 400, right m2; DN 432, right m1 (associated with DN 400); DN 438, right mandible with m1-m3; DN 590, left M1 or M2; DN 710, right mandible with deciduous p3; DN 857, left mandible with m3; DN 910, right mandible with m1-m3; DN 965, right mandible with deciduous p3; DN 1008, right mandible with deciduous p3; DN 1046, left M1 or M2; DN 4773, left m2.

Description. The 12 mandibular specimens attributed here to Oreotragus are derived from at least nine individuals (five adults with occluded third molars, four immature individuals retaining their deciduous premolars). The more complete DN 438 (m2 MD: 12.0 mm, BL: 5.8 mm; m3 MD: 16.0 mm, BL: 5.7 mm; Fig. 7C), DN 857 (m1 MD: 9.5 mm, BL: 6.2 mm; m2 MD: 11.8 mm, BL: 6.4 mm; m3 MD: 19.3 mm, BL: 6.5 mm) and DN 910 (m1 BL: 5.8 mm; m2 MD: 13.3 mm, BL: 6.5 mm; m3 MD: 15.4 mm, BL: 6.5 mm) specimens preserve relatively complete corpora and molars. The dental measurements of the DN Oreotragus specimens place them roughly in the narrow zone of overlap between the smaller-bodied fossil Oreotragus populations from Gondolin GD 2 and the larger-bodied Oreotragus from Haasgat HGD and Makapansgat Member 3 (see Adams, 2012b for comparative Oreotragus dental metrics). As noted in Adams (2012a) a comprehensive revision of the fossil record of the genus is essential to establish the significance of the metric variability in Oreotragus across South African Plio-Pleistocene deposits.

Tribe REDUNCINI Knottnerus-Meyer, 1907	
Genus REDUNCA C.H. Smith, 1827	
Type species Redunca redunca Pallas, 1767	
Redunca cf. fulvorufula (Afzelius, 1815)	

Referred specimens. DN 111, right partial horn core and orbital margin; DN 2573, right partial horn core; DN 4775, right partial horn core.

Description. The most complete of the reduncin horn core specimens attributed to Redunca cf. fulvorufula is DN 111, which preserves the right horn core pedicle as well as part of the right superior orbital margin and cranial vault. The two other right horn cores, while retaining less adhering cranial vault portions preserve the same core morphology as DN 111. The angle of horn core insertion, absence of sinuses in the pedicle, and development of the postcornual fossa are consistent with members of the tribe, and the size, insertion angle and cross-sectional shape of the core excludes attribution to genus Kobus Smith, 1840. Among extant and extinct Redunca species, the DN specimens are smaller than, and exhibit more upright and rounder horn cores, than both the extinct Redunca darti Wells and Cooke, 1956 from Makapansgat Member 3 (e.g., M 446 [type specimen], M 453, M 461, M 464, M 783) and extant Redunca arundinum Boddaert, 1785 comparative specimens (see also discussion of Redunca horn cores in Adams, 2006). The DN horn cores are similar in these morphological features to extant Redunca fulvorufula and the Gondolin GD 2 Redunca sp. horn cores (Adams & Conroy, 2005; Adams, 2006), but are from distinctly smaller-bodied individuals than the population sampled in the Gondolin GD 2 assemblage. Although the evolutionary relationships of South African Plio-Pleistocene Redunca and the diversity of reduncins during the Neogene remain unresolved (see Adams & Conroy, 2005; Adams, 2006; Adams, 2012a; Adams et al., 2010; Gentry, 2010), the morphology expressed by the Main Quarry specimens is most comparable with extant R. fulvorufula; a species present in the Cradle by at least the mid-Pleistocene (e.g., Gladysvale External Deposits; Lacruz et al., 2002).

Tribe TRAGELAPHINI Blyth, 1863	
Genus TRAGELAPHUS de Blainville, 1816	
Type species Tragelaphus scriptus Pallas, 1766	
Tragelaphus sp.	

Referred specimens. DN 120, right mandible with p2-m3; DN 163, left m3; DN 164, left m1; DN 165, left m2; DN 399, right p3; DN 1011, left M1; DN 1027, associated right p3, p4, and m1-m2 in a mandibular corpus; DN 2788, left maxilla with deciduous P4 – M2; DN 4112, right m1 or m2.

Description. The Drimolen Main Quarry Tragelaphus sample includes at least two different species. At least one species of large Tragelaphus is represented by the DN 399 and DN 2788 specimens. The crown of the DN 399 p3 is complete and unworn, but the roots are incompletely formed. The crown is brachydont and the morphology of the paraconid, metaconid and entoconid are consistent with extant Tragelaphus strepsiceros Pallas, 1766 comparative specimens. The DN 2788 maxilla retains the deciduous p4 and the M2 is just reaching the occlusal plane. Like the DN 399 specimen, the brachydonty and loph morphology is consistent with the genus, and size of the dentition is within the range of extant T. strepsiceros. Although these two specimens could potentially be derived from the same immature individual, we consider them as representing minimally two different individuals based on provenance: the DN 399 specimen was recovered from decalcified sediments (−2.3 m below datum; MBD) whereas the DN 2788 specimen was mechanically recovered from a surface-recovered ex situ calcified sediment block.

The remaining specimens are derived from at least one smaller species of Tragelaphus, likely representing minimally four individuals of Tragelaphus pricei Wells and Cooke, 1956 or the Tragelaphus scriptus lineage; however, lacking horn cores, we cannot provide a more specific attribution. The most complete of these specimens are the DN 120 and DN 1027 right mandibles (Fig. 6C). Both specimens exhibit similar dental metrics, corpus depths, premolar morphology (including paraconid-metaconid fusion on the p4) and buccal lophid rounding to the T. pricei holotype (M 18) and paratype (M 17, M 19) specimens from the Makapansgat Member 3 deposits (Wells and Cooke, 1956). The evolutionary relationships between the South African T. pricei and other smaller fossil Tragelaphus (e.g., Tragelaphus nkondoensis Geraads and Thomas, 1994) and extant T. scriptus is not resolved (see Gentry, 2010), and further assessment of the African record may eventually permit a more specific diagnosis of the Main Quarry specimens.

Tragelaphini gen. et sp. indet.	

Referred specimens. DN 1193, right m1 or m2 with associated tooth fragment in matrix.

Description. Two associated craniodental specimens (catalogued as DN 1193) are morphologically consistent with a large-sized tragelaphin but are not attributable to generic level. The associated tooth fragment is a partial molar (probably a lower buccal loph), while the more complete tooth is a damaged right lower molar lacking the lingual enamel surface. The angled buccal lophids and brachydonty of the lower molar is consistent with the tribe, and the size places the specimen within the range of extant Tragelaphus strepsiceros and smaller Taurotragus oryx individuals.

Bovidae gen. et sp. indet.	

Referred specimens. 538 specimens (see Table S1).

Description. The majority of the indeterminate bovid specimens from the Main Quarry deposits are postcranial elements (NISP = 365), with only 173 craniodental specimens (primarily nondiagnostic enamel fragments, mandibular incisors or poorly preserved horn core fragments) not attributed below the Family level (Table S1). Although these remains are undiagnostic, we would note that there is no indication of additional bovid taxa in the Main Quarry deposits beyond those established by the other referred specimens above. The one exception is the DN 647 partial left maxillary molar, which exhibits hypsodonty consistent with alcelaphins, but also resembles some of the smaller ovibovin remains potentially attributable to Makapania Wells and Cooke, 1956 from Swartkrans Member 1, Gladysvale, and Haasgat HGD (Vrba, 1976; Lacruz et al., 2002; Adams, 2012b).

Family GIRAFFIDAE Gray, 1821	
Giraffidae gen. et sp. indet.	

Referred material. DN 1097, right humerus.

Description. The DN 1097 specimen is an extremely large right cetartiodactyl humerus that preserves the medial portion of the condyle, supracondylar region, and the diaphysis to the distal-most portion of the humeral crest (Fig. 8A). Although no standard metrics could be gathered, the preserved length of the specimen is ∼26 cm and the length of the medial epicondyle to midsagittal of the condyle is ∼6.5 cm, indicating an approximate epicondylar width of ∼12–13 cm. In addition to the size of the preserved specimen, several morphological features align DN 1097 to extant giraffid comparative specimens (e.g., Giraffa camelopardalis Linnaeus, 1758) to the exclusion of a large bovid (e.g., extant Syncerus caffer Sparrman, 1779). The supracondylar diaphysis is ovoid (anteroposteriorly compressed, mediolaterally wide) rather than triangular from a more robust humeral crest. The medial articular surface of the condyle is anteroposteriorly flattened and cylindrical, and lacks a strongly developed sagittal ridge laterally.

Figure 8 Order Cetartiodactyla specimens attributed to the Families Giraffidae and Suidae from the Drimolen Main Quarry.

(A) DN 1097, Giraffidae gen. et sp. indet. right humerus, anterior (left) and posterior (right) views. (B) DN 2850, Suidae gen. et sp. indet. right third metatarsal, medial (left) and lateral (right) views. Scale bars equal 1 cm.

Unfortunately, there is a paucity of fossil giraffid postcranial from the South African palaeokarstic deposits, with only small collections described from Makapansgat Member 3 and the Swartkrans Members 1–3 deposits (Reed, 1996; Watson, 1993). Only a single humerus attributed to the extinct Sivatherium Falconer and Cautley, 1836 has been described from Swartkrans Member 1 Hanging Remnant (SK 3172), but this left humerus preserves only the non-overlapping proximal region. The penecontemporaneous eastern African record is more substantial, with postcranial samples from Koobi Fora, Olduvai, Omo-Shungura and Laetoli (Harris, 1976; Leakey & Harris, 1987; Harris, 1991; Robinson, 2011). The features that would allow for confident attribution of the DN specimen as either giraffine or sivatherine (e.g., elongation of the diaphysis, relative compression and width of the distal epiphyses) cannot be ascertained in the preserved state; and the humeri of Plio-Pleistocene giraffid species overlap significantly in simple linear metrics (Leakey & Harris, 1987; Harris, 1991; Robinson, 2011). Given these limitations, we treat this specimen as indeterminate below the Family level.

Family SUIDAE Gray 1821	
Suidae gen. et sp. indet.	

Referred material. DN 2850, right third metatarsal.

Description. The DN 2850 specimen is a nearly complete right third metatarsal from a juvenile suid, preserving a notably immature proximal articular surface (missing part of the medial articular surface) and an unfused distal metaphyseal surface (Fig. 8B). The diaphysis of the element is visibly robust (ML: 12.1 mm, dorsoventral depth: 10.1 mm) given its overall length (53.2 mm). Metric comparisons of the proximal articular and distal metaphyseal surfaces of DN 2850 against the third metatarsals of extant Phacochoerus aethiopicus Pallas, 1766 and Potamochoerus porcus Linnaeus, 1758 place this very immature element outside or just within the measured ranges of even fully adult P. aethiopicus and P. porcus individuals with complete proximal articular surface development and distal metaphyseal fusion (Table 6). The size of the specimen at its developmental stage would therefore seemingly preclude attribution to either of these extant genera.

Table 5 Linear dimensions (mm) of the Drimolen Main Quarry Antidorcas recki horn cores.

DN specimen	Anteroposterior depth	Mediolateral width	
Male	
224	43.6	28.1	
890	40.5	28.0	
1060	42.2	28.9	
3294	35.7		
Female	
995	17.5	14.6	
2483a	18.2	16.1	
4438	17.9	15.2	

Comparable fossil suid third metatarsals are extremely rare within African Plio-Pleistocene deposits (Bishop, 1994); however, a partial right third metatarsal of Metridiochoerus andrewsi (G 8105) has been described from the Gondolin GD 2 assemblage (Adams, 2006). The G 8105 specimen only preserves the proximal articular surface and part of the diaphysis and is derived from a more developmentally mature individual (or full adult), and the single comparable measurement reflects the larger body size of the species compared to the extant suids (Table 6). While the overall shape of the proximal articular surface and robusticity of the diaphysis appears shared between DN 2850 and G 8105, the significant ontogenetic differences prohibit confident attribution of the Main Quarry specimen to the same genus or species based solely on these broad similarities. Similarly, while other suid genera (e.g., Notochoerus, Kolpochoerus) are less frequently recovered than remains of Metridiochoerus in South African Plio-Pleistocene deposits, we also lack equivalent postcrania for these lineages to allow for direct comparisons. At present, we can only state that this specimen is derived from an extinct suid lineage that exhibited larger adult body size than extant or attributed fossil Phacochoerus or Potamochoerus.

Order PERISSODACTYLA Owen, 1848	
Family EQUIDAE Gray, 1821	
Genus EQUUS Linnaeus, 1758	
Type species Equus caballus Linnaeus, 1758	
Equus cf. quagga ssp. (Boddaert, 1785) (sensu Klingel, 2013)	

Referred material. DN 3424, left distal tibia; DN 4525, intermediate phalanx; DN 4781, partial ungual phalanx.

Description. The DN 3424 specimen preserves part of the distal metaphysis and articular surface from a left tibia, from the medial malleolus to the midline. There is no indication of persistence of a metaphyseal line indicating full closure of the distal growth plate and skeletal maturity of the element. Unfortunately, no standard metrics could be taken from the specimen to facilitate comparisons, but the element is directly comparable to extant E. quagga ssp. tibiae in size and morphology and is visibly smaller than the Equus capensis Broom, 1909 distal tibiae previously described from Swartkrans Members 1 Lower Bank (SKX 9596) and 2 (SKX 2390) (Churcher & Watson, 1993), as well as the Equus sp. specimen from the Haasgat HGD deposits (HGD 1015; Adams, 2012b).

The DN 4525 intermediate phalanx preserves the proximal epiphysis (ML: 45.9 mm, dorsoventral depth: 33.0 mm) and most of the diaphysis, but lacks the distal articular surface and some of the anterior face of the distal metaphyseal/diaphyseal region. The articular surfaces as preserved appear fully mature, there is no indication of a persistent metaphyseal line. As is the case with the DN 3424 tibia, the intermediate phalanx is consistent with extant E. quagga ssp. intermediate phalanges in size and morphology and smaller than previously described E. capensis specimens from Swartkrans Member 3 (SKX 39182: proximal ML: 50.0 mm [min.], proximal dorsoventral depth: 36.0 mm [min.]) and Haasgat HGD (HGD 1099, proximal ML: 53.5 mm, proximal dorsoventral depth: 33.8 mm) (Churcher & Watson, 1993; Adams, 2012b).

The DN 4781 specimen preserves only a partial proximal articular facet and part of the volar surface of an equid ungual phalanx. The specimen is visibly larger than the hipparionin ungual phalanges previously described from Swartkrans Member 1 (SKX 9166), Member 2 (SKX 2626), and Gondolin GD 2 (G 4218) (Churcher & Watson, 1993; Adams, 2006), and there is no indication of multiple nutrient foramina on the volar surface or posterior projection of the articular surface consistent with that tribe. The specimen appears to fall within the size range of extant E. quagga ssp. comparative specimens, and was recovered in close spatial proximity to the DN 3424 and 4525 specimens (see below). Collectively, this supports allocating this specimen to both E. quagga ssp. and potentially to the same individual as the other equid remains from the Main Quarry decalcified deposits.

Order PROBOSCIDEA Illiger, 1811	
Family ELEPHANTIDAE Gray, 1821	
Elephantidae gen. et sp. indet.	

Referred material. DN 4335, maxillary incisor fragment.

Description. A single tusk fragment (DN 4335) recovered from the decalcified sediments can be attributed to an indeterminate genus and species of elephantid (Fig. 9A). The recovery of this specimen, while relatively rare within other palaeokarstic deposits from the region, is not unexpected given the presence of minimally two separate elephantid tusk portions within the unexcavated calcified sediments of the Drimolen Main Quarry. The fragment exhibits both surface pitting and break-edge rounding, and it is uncertain whether the specimen preserves the most external, cortical enamel lamina or a more internal layer of enamel. The inner surface of the fragment exhibits fresh exposure of enamel that highlights the light longitudinal banding of the enamel, which under magnification is matched on the external, pitted surface of the specimen. The DN specimen is similar in laminar enamel thickness and suggested cross-sectional area and shape to the Swartkrans Member 1 Lower Bank specimens that have been previously attributed to Elephas sp. Linnaeus, 1758 (SKX 45691, 45692b; Watson, 1993); however, as a single tusk fragment we remain conservative in our attribution until further remains are recovered.

Figure 9 Specimens attributed to the Orders Proboscidea, Rodentia, and Tubulidentata from the Drimolen Main Quarry.

(A) DN 4335, Elephantidae gen. et sp. indet., maxillary tusk fragment, internal view. (B) DN 2760 Hystrix sp. left maxillary molar, occlusal view. (C) DN 1062 Orycteropus cf. afer right proximal radius, posteromedial view. Scale bars equal 1 cm.

Order HYRACOIDEA Huxley, 1869	
Family PROCAVIIDAE Thomas, 1892	
Genus PROCAVIA Storr, 1780	
Type species Procavia capensis Storr, 1780	
Procavia sp.	

Referred material. DN 552, left m1 or m2; DN 2365b, right I1; DN 2971, right and left I1 (antimeric); DN 3072, right distal femur; DN 4219, right I1.

Description. In contrast to the faunal samples recovered from most other South African palaeokarstic localities and deposits there is only an extremely small sample of hyrax remains at the Drimolen Main Quarry deposits (see Churcher, 1956; Brain, 1980). The collection of six total specimens are largely isolated central incisors. The antimeric pair (DN 2971) are most likely from a male individual given the sharp, centrally-positioned keel, while the DN 2365b and 4219 specimens exhibit the more rounded contour and mesial position of the central keel consistent with being from females (Churcher, 1956). None of the recovered teeth are metrically consistent with Procavia transvaalensis Shaw, 1937 (see comparative measurements in Churcher, 1956). Similarly, the DN 3072 distal femur is homologous to extant Procavia capensis Pallas, 1766 comparative specimens and visibly smaller than the P. transvaalensis distal femur from Haasgat HGT (HGT 1004; Adams, 2012b). As the composition of the sample does not allow us to diagnose whether the sample is derived from the extinct Procavia antiqua Broom, 1934 (see also Schwartz, 1997) and/or extant P. capensis, we can only attribute the Main Quarry specimens to a species of Procavia that is not P. transvaalensis.

Order LAGOMORPHA Brandt, 1855	
Family LEPORIDAE Fischer von Waldheim, 1817	
Genus PRONOLAGUS Lyon, 1904	
Type species Pronolagus crassicaudatus Geoffroy, 1832	
Pronolagus sp.	

Referred material. DN 2823, right mandible preserving the p3-m2.

Description. Only one of the 38 identified leporid remains could be identified as representing as species of Pronolagus in the Drimolen Main Quarry deposits. The overall size of the specimen is significant smaller than in extant Lepus capensis Linnaeus, 1758, and the p3 exhibiting both anterior reentrants and no evidence of the posteroexternal reentrant extending nearly to the lingual border of the tooth as in Lepus. Amongst extant analogues, the overall mandibular morphology is similar to Pronolagus rupestris Smith, 1834 but the dentition and corpus is somewhat smaller than both male and female P. rupestris comparatives. Given the currently limited data on the South African leporid record (Winkler & Avery, 2010), we retain the specimen at the generic level pending a more comprehensive review of the South African Plio-Pleistocene lagomorphs.

Leporidae gen. et sp. indet.	

Referred material. DN 771, right immature proximal tibia; DN 1083, left calcaneus; DN 2104, right calcaneus; DN 2155, left distal scapula; DN 2256, left immature proximal tibia; DN 2330, left mandibular corpus; DN 2333, right maxillary premolar or molar; DN 2341, left calcaneus; DN 2342, left calcaneus; DN 2365c, left distal humerus; DN 2571, right immature distal femoral epiphysis; DN 2715a, right distal humerus; DN 2715b, left proximal ulna; DN 2736, right immature calcaneus; DN 2798, right proximal tibia; DN 2800, right immature distal femoral epiphysis; DN 2805, right immature distal femoral metaphysis (articulates with DN 2800); DN 3059, left immature distal femoral epiphysis; DN 3304, left distal scapula; DN 4114, left proximal ulna; DN 4116, left distal humerus; DN 4118, left calcaneus; DN 4120, associated set of three lumbar vertebrae and sacrum; DN 4127, right immature proximal ulnar metaphysis; DN 4208, right distal humerus; DN 4225, left distal humeral metaphysis; DN 4226, left distal tibia; DN 4227, right distal scapula; DN 4228, left proximal ulna; DN 4229, partial edentulous maxilla; DN 4257, left immature calcaneus; DN 4381, left immature distal femoral epiphysis; DN 4399, indet. side immature distal femoral epiphysis; DN 4531, left proximal tibial metaphysis.

Description. The indeterminate leporid sample includes 34 numbered specimens (representing 36 total elements). Other than fragmentary and indeterminate craniodental remains (DN 2333 and DN 4229), the current collection is comprised of postcranial remains from what are likely two different species. The majority of the remains are derived from a lagomorph somewhere in body size between extant Lepus capensis and Pronolagus rupestris. A smaller lagomorph that had adult and immature postcranial elements smaller than adult extant P. rupestris is represented by DN 2104, 2155, 2715a & b, 2800, 2805, 3059, 4120, and 4228.

Order RODENTIA Bowdich, 1821	
Family HYSTRICIDAE Fischer de Waldheim, 1817	
Genus HYSTRIX Linnaeus, 1758	
Type species Hystrix cristata Linnaeus, 1758	
Hystrix sp.	

Referred material. DN 2760, left maxillary molar (M2?).

Description. Only a single identifiable left maxillary molar can be attributed to an indeterminate species of porcupine (Fig. 9B). The presence of both mesial and distal interstitial wear facets indicate that the specimen is either an M1 or M2; we favour the latter based on both the size and occlusal outline but cannot confidently attribute the position. Metrically, the specimen (MD: 8.2 mm, BL: 9.3 mm [min. given minor enamel flaking]) falls within the range of extant Hystrix africaeaustralis Peters, 1852 maxillary M1s and M2s and is smaller than both Hystrix makapanensis Greenwood, 1958 and Xenohystrix crassidens Greenwood, 1955 (Adams, 2012a). We would note, however, that the current metric range for H. makapanensis M1s and M2s is based on an extremely small sample from the Gondolin GD 2 assemblage (e.g., derived from two individuals; Adams, 2012a). As an isolated tooth lacking other diagnostic features we elect to retain the specimen at the generic level.

Order TUBULIDENTATA Huxley, 1872	
Family ORYCTEROPODIDAE Gray, 1821	
Genus ORYCTEROPUS Geoffroy Saint-Hilaire, 1796	
Type species Orycteropus afer Pallas, 1766	
Orycteropus cf. afer	

Referred material. DN 1062, right proximal radius.

Description. The DN 1062 right proximal radius preserves a nearly complete head with articular surfaces and radial tuberosity, but very little diaphysis distal to the tuberosity (Fig. 9C). Although there is minor abrasion damage to the cortex around the radial head, the minimum linear dimensions are 17.1 mm mediolaterally and 12.5 mm dorsoventrally. The articular surface morphology of the specimen is diagnostically orycteropodid (to the exclusion of similarly sized carnivores, primates, Hystrix and Smutsia [Order Philodota]) in the well-developed medial articular facets for contact with the lateral coronoid process of the ulna (Fig. 9C). The DN specimen is identical to extant Orycteropus afer radii available to us for direct comparison in both morphology and size, and exhibits excellent joint congruence with the capitulum of the Orycteropus cf. afer distal humerus from Swartkrans Member 1 (SKX 14261) (Lehmann, 2004).

Biochronology

Fauna from the site has been recovered from both calcified and decalcified deposits in the Main Quarry using a variety of techniques over the years. A small portion of the collections from the early 1990s was developed from blocks recovered from the lime miners dumps just outside the Main Quarry. While these blocks have always been assumed to come from the Main Quarry deposits due to their proximity, the recent excavation of a newer, older (∼2.6 Ma) 50 m west of the Main Quarry (the Drimolen Makondo; Rovinsky et al., 2015; A Herries et al., 2016, unpublished data) does mean that this ex situ material could be mixed from more than one age of deposit. However, these two different deposits are not stratigraphically intertwined as is the case for deposits like Members 4 and 5 at Sterkfontein (Herries & Shaw, 2011) or Swartkrans Members 1–3 (Herries & Adams, 2013). This is currently no indication that similarly-aged Pliocene deposits exist in the Main Quarry itself.

Although there is notable diversity in the recovered Drimolen Main Quarry faunas, the majority of the specimens provide limited biochronological data for interpreting the age of the deposits. As is typical for South African palaeokarstic deposits in the region, the recovery of Equus cf. quagga ssp. from the in situ deposits indicates that at least part of the assemblage formed after 2.33 Ma given the first appearance of the genus in African early Pleistocene deposits (Geraads, Raynal & Eisenmann, 2004). The remainder of the ungulate specimens identified from the Main Quarry are either not specifically attributable, are members of long-surviving Plio-Pleistocene lineages, or have poorly-secured first appearance (FAD) and last appearance dates (LAD) in South Africa. The largest bovid sample in the assemblage, Antidorcas recki, has been recovered from eastern and South African deposits spanning the late Pliocene (e.g., Shungura Formation B-H; Gentry, 2010; McDougall et al., 2011; Brown, McDougall & Gathogo, 2013) to the >1.07 Ma (or <0.780 Ma) Olduvai Bed IV (Tamrat et al., 1995) and the 1.07–0.780 Ma Elandsfontein deposits (Gentry, 2010; Braun et al., 2013). The differences in the transverse ridging of the horn cores with the previously described Bolt’s Farm Pit 3 specimens may or may not reflect a temporal difference in depositional age.

Although this analysis has not addressed the non-hominin primates from the Drimolen Main Quarry, data reported in Nieuwoudt (2014) and reproduced in Table 1 suggests a fairly homogenous sample of cercopithecoids relative to other nearby early Pleistocene palaeokarstic deposits that typically record greater taxonomic diversity (Brain, 1980; Jablonski & Frost, 2010). The specifically attributable DMQ specimens have been allocated to either the extinct papionin subspecies Papio hamadryas robinsoni Freedman, 1957 (alt. Papio robinsoni; Gilbert et al., 2015) or the extinct colobine species Cercopithecoides williamsi Mollett, 1947. Both of these species have FADs in the late Pliocene (a constrained FAD for Papio somewhat less clear, see Jablonski & Frost, 2010; Gilbert et al., 2015) and are common in post ∼2 Ma early (potential to mid-) Pleistocene South African deposits (e.g., Swartkrans, Sterkfontein Member 5, Kromdraai, Cooper’s D, Gladysvale; Jablonski & Frost, 2010). As such, neither primate provides strong biochronological constraints on the depositional age of the Main Quarry deposits beyond being consistent with species recovered from other nearby early Pleistocene localities (e.g., Sterkfontein Members 4 and 5, Swartkrans Members 1–3, Kromdraai A and B) (Brain, 1980; Jablonski & Frost, 2010).

The carnivoran specimens provide a more constrained depositional age. Remains of the genus Chasmaporthetes have been recovered across African localities, with C. nitidula described from South African deposits ranging from Sterkfontein Jacovec Cavern and Members 2 and 4 (<2.46–2.01 Ma; Herries et al., 2013) to as late as Swartkrans Member 3 (sometime between 1.3 and 0.6 Ma; Herries, Curnoe & Adams, 2009). However, Lycyaenops silberbergi has been recovered from a far narrower range of terminal Pliocene and early Pleistocene deposits of Laetoli (as Lycyaenops cf. L. silberbergi; Werdelin & Dehghani, 2011) in East Africa and Sterkfontein in South Africa (∼3.8–2.02 Ma; Turner, 1990; Turner, 1997; Werdelin & Lewis, 2005; Werdelin & Peigné, 2010; Herries & Shaw, 2011; Herries & Adams, 2013). A single mandibular specimen (SK 300) of Lycyaenops silberbergi has been described from Swartkrans Member 1 (Ewer, 1955b), and although questions over provenience has been raised it is still considered derived from these deposits (see discussion in Turner, 1987b); this effectively establishes an LAD for the species within these deposits of 1.96-1.80 Ma (Pickering et al., 2011a; Herries & Adams, 2013).

The identification of Dinofelis cf. barlowi from the DMQ also argues for a late Pliocene-early Pleistocene deposition. This apparently endemic South African machairodont has been definitively recovered from a temporally narrow range of deposits spanning approximately 2.7–1.98 Ma (Werdelin & Lewis, 2001; Lacruz, Turner & Berger, 2006; Herries, Curnoe & Adams, 2009; Werdelin & Peigné, 2010; Herries & Shaw, 2011; Pickering et al., 2011b; Kuhn et al., 2012). What cannot be determined at this point is the biological relationship of this specimen to the previously described Dinofelis aff. piveteaui specimens, as no maxillary canine was recovered or described in association with those craniodental remains. As discussed by O’Regan & Menter (2009), the basis for classifying the DMQ Dinofelis remains as D. aff. piveteaui is based largely on P4 morphology. The protocone of the DN 1012 P4 is reduced; not as reduced as D. piveteaui specimens from Kromdraai A or Motsetse but yielding a buccolingually-narrower tooth than is present in D. barlowi. Simultaneously, the P4 metastyle blade is only slightly mesiodistally elongated relative to D. barlowi and lacks the elongation exhibited by South African D. piveteaui P4 specimens. As with much of the anatomy of fossil taxa established from small sample sizes, it is difficult at present to establish whether these P4 features reflect sexual dimorphism, individual, or chronological variation within either D. barlowi or D. piveteaui. And while further craniodental specimens could ultimately link the DN 2791 canine with these other DMQ Dinofelis remains, there is no morphological affinity to D. piveteaui maxillary canines to support treating the specimen as derived from the same population.

Ultimately, if these previously described remains cannot be accommodated with either species sensu stricto we posit four potential interpretations of the DMQ Dinofelis material assigned to D. aff. piveteaui by O’Regan & Menter (2009) that will require further analysis to test whether: (1) they represent a derived D. barlowi; (2) they represent a primitive D. piveteaui sensu stricto; (3) they are derived from a discrete population undergoing anagenesis from a more primitive Dinofelis (e.g., D. barlowi (O’Regan & Menter, 2009) or Dinofelis aronoki (Werdelin & Lewis, 2001)) to D. piveteaui in the early Pleistocene; or (4) they represent a separate, previously unrecovered species of Dinofelis in the African record (which we view as less likely). If the DMQ Dinofelis aff. piveteaui remains represent a novel species, they would ultimately contribute little to interpreting deposit biochronology. As D. barlowi does not occur in South African deposits after 1.98 Ma, if the DMQ Dinofelis aff. piveteaui remains are a derived D. barlowi they would likely indicate deposition after 1.98 Ma—but would similarly lack an established FAD or LAD. Finally, given the recovery of the D. piveteaui type specimen from the (tentatively dated) 1.89–1.63 Ma Kromdraai A deposits (Herries, Curnoe & Adams, 2009; Gilbert, Frost & Delson, 2016) and in east Africa by 1.61 Ma (Werdelin & Lewis, 2005; McDougall et al., 2011), then the DMQ Dinofelis aff. piveteaui specimens would suggest a depositional date prior to ∼1.6 Ma and/or the Kromdraai A assemblage.

In sum, the DMQ faunal assemblage includes species only occupying the South African landscape after 2.33 Ma (Equus cf. burchelli ssp.), and includes the remains of carnivore species whose currently reconstructed LADs within the early Pleistocene indicate that at least some of the deposits formed prior to (or near) 2.02 Ma (Lycyaenops silberbergi) to 1.98 Ma (Dinofelis cf. barlowi). At present, the biochronological significance of the Drimolen Main Quarry D. aff. piveteaui is complicated by an unreliable FAD and LAD, but may suggest some deposition after 1.98 to potentially 1.89-1.6 Ma.

Discussion

This first description and analysis of the macromammalian fossil specimens from the Drimolen Main Quarry assemblage has documented remarkably high taxonomic diversity (9 Orders, 14 Families) relative to sample size (NISP: 1390, MNI: 147). While deposits like Swartkrans Member 1 record a somewhat greater range of taxa (9 Orders, 17 Families), the number of non-hominin taxonomically identifiable specimens underlying this diversity is substantially higher (NISP: 4583, MNI: 103; Watson, 1993). Despite this diversity, the McIntosh evenness statistic for the DMQ is low (0.71) and one standard deviation below the mean (0.80; range 0.60–0.93; s.d. 0.09) for South African early-mid Pleistocene localities (Table 7) and close to the value of the Haasgat HGD ex situ assemblage (0.72; Adams, 2012b). This low value reflects the high proportion of Papio hamadryas robinsoni individuals relative to all other taxa recovered from the DMQ deposits. With the inclusion of Cercopithecoides williamsi and indeterminate specimens, 38.4% of the DMQ assemblage come from the Family Cercopithecidae; equal to the proportion (38.4%) of the second largest taxonomic group, the Family Bovidae. In this respect, DMQ is also similar to Haasgat HGD where non-human primates make up 35.1% and bovids make up 40.9% of the assemblage (Adams, 2012b); although we would reinforce that we are only considering the non-hominin components of the DMQ assemblage in comparison to the Haasgat HGD assemblage that lacks hominin remains.

Table 6 Comparative measurements (mm) of the DN 2850 Metridiochoerus sp. right immature third metatarsal and fossil and extant adult suid third metatarsals.

			Phacochoerus aethiopicus	Potamochoerus porcus	
Measurement	DN 2850	G 8105	Mean	Min.	Max.	n	Mean	Min.	Max.	n	
Proximal dorsoventral depth	22.0	30.0	19.6	18.1	20.9	6	22.1	21.3	23.3	4	
Distal mediolateral width	16.2	–	15.4	13.7	16.9	5	15.7	14.3	16.9	4	
Distal dorsoventral depth	17.0	–	15.5	14.6	16.5	6	16.5	15.7	17.7	4	

Table 7 McIntosh evenness statistic values for the Drimolen Main Quarry and comparative South African fossil assemblages.

Sitea	Evenness	
Drimolen Main Quarry	0.71	
GD 2	0.60	
Gladysvale	0.92	
Haasgat HGD	0.72	
Kromdraai A	0.83	
Kromdraai B	0.76	
Makapansgat 2	0.93	
Makapansgat 3	0.81	
Makapansgat 4	0.84	
Makapansgat 5	0.78	
Sterkfontein Member 4	0.67	
Sterkfontein 53 Breccia	0.86	
Sterkfontein Olduwan Infill	0.74	
Sterkfontein Member 5 West	0.84	
Swartkrans Member 1 Lower Bank	0.83	
Swartkrans Member 2	0.86	
Swartkrans Member 3	0.84	
Notes.

a Data from sources listed in Adams (2006) and Adams (2010) but modified to exclude hominin taxa from the calculation.

The composition of the Drimolen Main Quarry bovid assemblage is not taxonomically unique relative to other penecontemporaneous South African fossil assemblages, but the numerical dominance of Antidorcas recki relative to other taxa is somewhat unusual (NISP: 25, MNI: 16, 28.6% total bovid assemblage). The only other sites with a similar sample size of A. recki are Cooper’s D (MNI: 12; de Ruiter et al., 2009) and Kromdraai A (NISP: 44, MNI: 13; Brain, 1980). In contrast to the DMQ assemblage, both have a far more substantial representation of alcelaphins (Cooper’s D: MNI: 46; Kromdraai A: NISP: 220, MNI: 51) and A. recki only represents 12.0% (Cooper’s D) and 11.5% (Kromdraai A) of the total bovid assemblage. What stands in rather strong contrast to the typical element-wise representation of bovid elements is the high frequency of horn cores in the DMQ assemblage (see Brain, 1980; Watson, 1993; Pickering, 1999), particularly in the case of A. recki, where very few teeth have been recovered alongside a substantial number of horn cores. Interpreting the origin and significance of this pattern of element preservation in the bovid sample compared to other palaeokarstic deposits is beyond the scope of the present paper, but will potentially be informative on the taphonomic processes underlying the Drimolen Main Quarry deposit formation.

The only other substantially represented taxonomic group in the DMQ assemblage is the Order Carnivora, which comprises 12.9% of the faunal assemblage and is biased towards the Family Felidae (57.9% of the Carnivora and 7.5% of the overall faunal assemblage). This positions the DMQ assemblage as particularly carnivore-rich, with only Kromdraai B (15.8%), Swartkrans Member 3 (16.9%) and the Sterkfontein Member 5 deposits (West: 30.7%; Olduwan Infill: 13.6%; STW 53 Breccia: 13.3%) having higher proportions of carnivores amongst the penecontemporaneous South African palaeokarstic assemblages. The proportion of carnivores in the DMQ assemblage is also substantially less than the recently described Drimolen Makondo deposits (25.0%; Rovinsky et al., 2015; A Herries et al., 2016, unpublished data), although the limited faunal sampling of the Makondo to-date (like other provisionally described faunal assemblages such as Motsetse (Berger & Lacruz, 2003), Hoogland (Adams et al., 2010), Malapa (Dirks et al., 2010; Kuhn et al., 2012)) necessitates further sampling to ensure these proportions are not artefacts of sample size. The co-occurrence of large predatory felids (with minimally three subfamilies and five genera represented in the deposit)—for example, Panthera, Dinofelis and Megantereon—is not uncommon in South African early Pleistocene deposits (Brain, 1980; Werdelin & Peigné, 2010). What is unusual is the potential sympatric occurrence of two species of Dinofelis recorded by the DMQ Dinofelis cf. barlowi and Dinofelis aff. piveteaui, specimens. While it has been noted that temporal overlap of two Dinofelis species may have been relatively common, at least in East Africa (e.g., Lewis & Werdelin, 2007), this current study would represent the first strong evidence of such contemporaneous overlap within a single deposit (Werdelin & Lewis, 2001; Werdelin & Lewis, 2013; Werdelin & Peigné, 2010).

The Drimolen Main Quarry faunal assemblage is also unique relative to most penecontemporaneous South African karstic deposits in the low representation of porcupine (Family Hystricidae) and hyrax (Family Procaviidae) remains in the deposits. These two families only comprise 2.7% of the total DMQ faunal assemblage, which falls below even the low proportions of these taxa at Sterkfontein Member 4 (4.2%; Brain, 1980) and Kromdraai A (9.0%; Brain, 1980), and strongly contrasts the representation of these families in the Swartkrans Member 1 (21.4%; Watson, 1993) and Haasgat HGD (19.5%; Adams, 2012b) faunal assemblages. There are a number of geologic (e.g., entrance morphology, deposit time-averaging), ecologic (e.g., immediate habitat and resource distribution) and/or taphonomic factors (e.g., pre-, peri- and postdepositional) that could underlie this low representation of these karst-utilising taxa that will require further analysis to assess.

In sum, this comprehensive accounting of the Drimolen Main Quarry faunal assemblage allows us to provide some initial comments on the palaeohabitats of the Drimolen region during depositon; although we note that a more comprehensive palaeoecological interpretation and analysis integrating stable isotopic results from the assemblage will be forthcoming. The overall vegetative communities and landscape ecology suggested by the taxon presence and abundance recovered from the DMQ deposits is largely consistent with the mixed, open-to-closed palaeohabitats that have been reconstructed for the other regional, penecontemporaneous South African palaeokarst deposits (Vrba, 1976; Vrba, 1995; Brain, 1980; Benefit & McCrossin, 1990; McKee, 1991; Rayner, Moon & Masters, 1993; Brain, 1995; Schmid, 2002; Elton, 2007; see also summaries in Reed, 1996; Kuman & Clarke, 2000; Adams, 2006). In part this reflects the broad habitat types that most of the specifically identifiable taxa from the DMQ deposits can occupy. Both previously attributed primate taxa are larger-bodied and terrestrial (Papio hamadryas robinsoni, Cercopithecoides williamsi) and may have been sympatric, niche-partitioned primates adapted to the progressively more open habitats of the early Pleistocene (Elton, 2007; Jablonski & Frost, 2010). However, the composition and resource availability of these more open habitats, and how they contrast with earlier palaeoecosystems that supported cercopithecoids like Parapapio (that disappear from the record in the early Pleistocene), remains unresolved. Even the more unusual species recovered from the Drimolen Main Quarry deposits (indeterminate elephant, giraffe, and aardvark [Orycteropus cf. afer]) can unfortunately provide few constraints on the palaeohabitat types, distribution or proportions near the site. Neither the elephant or giraffe individual specimens could be confidently attributed below the level of Family, and extant aardvark populations occupy highly variable habitats across Africa with the only limitation being access to social insects (Kingdon & Hoffmann, 2013b).

The dominance of Antidorcas recki in the bovid sample does not provide any specific evidence for reconstructing the overall vegetative community structure around Drimolen without further isotopic analysis. Although prior analyses of Sterkfontein and Swartkrans specimens have reconstructed the diet of this antilopin as a browser (Lee-Thorp & Van der Merwe, 1993; Lee-Thorp, Van der Merwe & Brain, 1994; Van der Merwe et al., 2003), a single specimen from Gondolin was interpreted as a mixed feeder (Adams, 2012b) and sampled individuals from Olduvai have documented greater dietary flexibility and an increase in graze in the species through the Bed I deposits associating them more with bushland to grassland palaeohabitats (Plummer et al., 2009). The only bovid taxa recovered from DMQ with specific habitat requirements are the mountain reedbuck (Redunca cf. fulvorufula) and klipspringer (Oreotragus sp.). Extant mountain reedbuck graze primarily in montane grasslands on rocky, hilly and/or broken terrain (Irby, 1976; Kingdon & Hoffmann, 2013b); klipspringer exhibit a suite of musculoskeletal, physiological and behavioural adaptations to browsing on pair-defended rock outcrops (Norton, 1980; Kingdon & Hoffmann, 2013b). The presence of some uplifted topography and/or local kopjes with montane grasses and browse near Drimolen during the formation of the Main Quarry assemblage is further reinforced by the recovery of hyrax (Procavia sp.) and rock hare (Pronolagus sp.) (Kingdon & Hoffmann, 2013b; Happold, 2013).

The extinct DMQ large carnivores are similarly reconstructed as adapted for hunting within a range of open and closed palaeohabitats. Both Dinofelis and Megantereon have been reconstructed as ambush predators preferring a closed-mixed habitat (Marean, 1989; Lewis, 1995a; Lewis, 1995b; Lewis, 1997; Palmqvist et al., 2003; Christiansen & Adolfssen, 2007; Palmqvist et al., 2008). In its postcranial anatomy, the smaller-bodied Megantereon is reminiscent of the extant jaguar (Panthera onca), with relatively short, heavily muscled, and powerful limbs (Lewis, 1995b; Lewis, 1997; Christiansen & Adolfssen, 2007). Isotopic analyses of European Megantereon support the preference for a closed-mixed habitat, suggesting that the genus predated on browsers and mixed-feeders (Palmqvist et al., 2003; Palmqvist et al., 2008). Dinofelis has been shown to have had greater flexibility in the forelimb than seen in the other African machairodonts (i.e., Homotherium, Megantereon; Lewis, 1995a; Lewis, 1995b; Lewis, 1997) and a trend in the larger-bodied species toward a more Panthera-like cranial and postcranial anatomy, suggesting that although it seems to have been an ambush grappler much like Megantereon it may have been able to exploit a larger range of habitats than the other African sabertooth genera (Lewis, 1997; Lewis & Werdelin, 2007). Although postcranial remains of African Lycyaenops and Chasmaporthetes species are extremely rare (see Rovinsky et al., 2015), cursorial adaptations within the hunting-hyaena lineage supports more open habitats in the region, in contrast to the felids (Khomenko, 1932; Galiano & Frailey, 1977; Berta, 1981; Tseng, Li & Wang, 2013). Amongst the extant carnivores identified by O’Regan & Menter (2009) modern populations of Cape fox (Vulpes chama), yellow mongoose (cf. Cynictis penicillata), and meerkat (aff. Suricata suricatta) typically occupy open, semi-arid to arid ecosystems (Kingdon & Hoffmann, 2013a). Interestingly, the extensive burrowing behaviour of meerkat and yellow mongoose also imply the presence of deep soils, with the latter avoiding rocky and hard soil regions for burrow construction (Kingdon & Hoffmann, 2013a).

Conclusions

This first description of the non-primate faunas of the Drimolen Main Quarry provides an insight into the macromammalian community structure of a previously undescribed part of the Cradle of Humankind during the Early Pleistocene of South Africa. The fauna recovered suggests a community of great taxonomic breadth. Amongst the sampled Order Carnivora in particular there is a surprising amount of diversity, with a minimum of four large species of felid present alongside at least four hyaenid species, including the first strong evidence of two species of Dinofelis recovered from a single deposit (but see Werdelin & Lewis, 2013). Although there is a large amount of taxonomic diversity amongst the fauna, non-hominin primates comprise a significant percentage of the assemblage—a disparity that will only increase when the substantial Drimolen hominin sample is included in the overall faunal picture. As this publication provides the primary description of the recovered DMQ faunas, further analyses will need to be undertaken to understand the idiosyncrasies of the site, ranging from the taphonomic processes shaping the assemblage (particularly the large number of bovid horn cores and concomitant paucity of dental and postcranial remains) to a more integrated palaeoecological analysis of the Main Quarry.

Supplemental Information

Table S1 Catalogue of the indeterminate bovid craniodental and postcranial remains from the Drimolen Main Quarry deposits

Click here for additional data file.

The Drimolen Main Quarry faunas described here have been developed over the course of over two decades, and we would like to acknowledge the efforts of past and ongoing excavation assistants, field school students and preparators in developing this material; specifically Stephanie E. Baker for her ongoing curatorial and site management work. We would also like to thank Bernhard Zipfel (Evolutionary Studies Institute, University of the Witwatersrand) for facilitating access to the DMQ fossil specimens, as well as Shaw Badenhorst (Archaeozoology and Large Mammals Collections), Stephany Potze and Lazarus Kgasi (Plio-Pleistocene Section) of the Ditsong National Museum of Natural History for facilitating access to modern and fossil comparative collections used in this research. Rhiannon Stammers assisted with the Dino-Lite Edge AM4815ZTZ microscope used to evaluate several of the DN specimens described here.

Additional Information and Declarations

Competing Interests

Author Contributions

Data Availability

The authors declare there are no competing interests.

Justin W. Adams and Douglass S. Rovinsky conceived and designed the experiments, performed the experiments, analyzed the data, contributed reagents/materials/analysis tools, wrote the paper, prepared figures and/or tables, reviewed drafts of the paper.

Andy I.R. Herries conceived and designed the experiments, contributed reagents/materials/analysis tools, wrote the paper, prepared figures and/or tables, reviewed drafts of the paper.

Colin G. Menter conceived and designed the experiments, contributed reagents/materials/analysis tools, wrote the paper, reviewed drafts of the paper, management and funding of fossil site; Excavation of fossil specimens.

The following information was supplied regarding data availability:

The research in this article did not generate any additional raw data beyond what is presented here. All Drimolen Main Quarry fossil specimens are curated at the Evolutionary Studies Institute, University of the Witwatersrand and accessible to researchers upon request.

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
