# Peer review of "Macromammalian faunas, biochronology and palaeoecology of the early Pleistocene Main Quarry hominin-bearing deposits of the Drimolen Palaeocave System, South Africa"

_PeerJ, doi:10.7717/peerj.1941_

## Round 0.1 · original submission · Minor Revisions

Although the reviewer suggestions are few and relatively minor, they still need to be considered in your revision prior to publication. The point about "precision" of measurement is well taken.

·

Basic reporting

The authors state in both the Introduction (lines 145-14) and in the Materials and Methods (lines 181-185) that they are not discussing the carnivores and primates from Drimolen. This is understandable given that they are published elsewhere, but we could probably cut out some of this duplication.

The authors occasionally present measurements to the hundredth of a millimeter (e.g. table 3 and throughout the text). I realize that this is a pet peeve of mine, but I seriously doubt that even Mitutoyo calipers are really reliable and replicable at the scale of tens of microns. For instance, as I am writing I see a measurement on line 998 of 17.06 mm for an Orycteropus radius; is this really 17.06 mm, or could it be 17.05 mm or even 17.04 mm? The difference of course is trivial, but I always doubt such instances of hyper-precise measurements.

In lines 1088-1104 the authors discuss 4 hypotheses relating to the identification of a Dinofelis specimen and its importance for the age estimate of the site. These hypotheses result in younger age estimates for the site ranging from 1.98 to 1.89-1.63 to 1.6 Ma. In the following paragraph the authors summarily state that the younger age estimate is 1.98 Ma without stating why this is the preferred hypothesis. More discussion is needed to support this age estimate.

The authors state the only DMQ and Kromdraai A have large samples of A. recki. However, Coopers D also has a large sample of A. recki fossils (12 individuals; de Ruiter et al., 2009 JHE 56: 497-513).

Otherwise, the paper is clearly written and well illustrated. The figures and tables are appropriate and necessary to document the assemblage.

Experimental design

The research design of this paper is appropriate for the material being discussed. The authors provide an excellent summary of the history of excavation of the cave, and the best discussion of the geological formation of the cave that I have yet seen.

The research question is clearly stated, relevant, and meaningful. The methods are easily replicable. The investigation was conducted rigorously and to a high technical standard.

Validity of the findings

The findings presented in this report are appropriate and valid, and the data are robust. The conclusions are appropriately stated, and are supported by the evidence presented throughout the paper. This paper presents a great deal of new information not available elsewhere, while also incorporating relevant prior publications on Drimolen.

Additional comments

Line 31-32: In the abstract, I would state the DNH 7 is the most complete skull of P. robustus ever found, not just cranium

Line 97: “that often grade . . . “

Line 536: it was my understanding that Vrba 1977 referred SK 3211b to Rabaticeras porrocornutus, not Numidocapra. Is there another reference that should be here?

Line 1014: “of a newer, older deposit . . . .”

Line 1099: …would likely indicate deposition before 1.98 Ma . . . “

Reviewer 2 ·

Basic reporting

This is a well-written report on the non-primate fauna at Drimolen. Introduction and Discussion include relevant literature. Figures and tables present relevant information. I have some small comments in the attached annotated PDF. One very minor detail is to be consistent throughout in the discussion of the non-hominid primates.....they are not really "published", as mentioned in the Materials and Methods. There are a couple of typos (spell-check insists on turning Koobi Fora into Koobi For a) that need to be corrected, but otherwise the MS is very clean.

Experimental design

Experimental Design is straightforward description and discussion of biochronological implications. This is done very well.

Validity of the findings

Findings appear valid based on the information and data presented.

Additional comments

This is a very nice article that requires little revision, in my opinion. A few minor comments are provided in an annotated PDF. However, I must admit that I am not an expert in Carnivores or Bovids, the main groups focused on in the article and perhaps the most important in terms of biochronology, so I would weight any suggested edits/comments from experts in these fields a bit more heavily than my own.

Annotated reviews are not available for download in order to protect the identity of reviewers who chose to remain anonymous.

---

## Round 0.2 · accepted · Accept

Thank you for your careful attention to reviewer comments and suggestions.